# Delving into Weakly Supervised Learning with Pre-Trained Models

## Abstract

Weakly supervised learning (WSL) is a popular machine learning paradigm in recent years that aims to learn a classifier with incomplete, imprecise, or inaccurate supervision. Existing WSL approaches have mainly focused on designing different loss functions or training strategies and then training models from scratch. In this paper, we first empirically show that a zero-shot baseline based on the Contrastive Language-Image Pre-Training (CLIP) model with class descriptions empowered by GPT-4o can outperform previous state-of-the-art methods trained from scratch on various WSL problems. Therefore, this motivates us to fine-tune pre-trained models to further improve the performance. However, our additional experiments show that naive use of existing WSL losses degrades performance due to severe overfitting exacerbation and feature degeneration problems. To address these problems, we propose a novel weakly supervised fine-tuning approach using dual classification heads that are trained synergistically by alternately distilling reliable supervision and performing efficient model fine-tuning. Theoretically, we prove the consistency and convergence rate of the proposed risk estimator. Empirically, extensive experiments on benchmark datasets of different WSL problems validate the effectiveness of the proposed approach against state-of-the-art competitors. The code is provided at
https://github.com/ICLR2025-6897/WSFT_code.

## 1 Introduction

The success of deep learning depends on a large amount of training data with high-quality labels. However, such a condition can be an obstacle in many real-world scenarios where labeled training data is scarce. Weakly supervised learning (WSL) aims to train a classifier with only weak supervision that can achieve comparable performance to fully supervised learning approaches (Zhou, 2018; Sugiyama et al., 2022). In recent years, WSL has made great progress, and a variety of WSL problems have been studied, such as positive-unlabeled (PU) learning (Elkan & Noto, 2008; Kiryo et al., 2017; Bekker & Davis, 2020), unlabeled-unlabeled (UU) learning (Lu et al., 2019; 2020; Xie et al., 2024), and similarity-based classification (Bao et al., 2018; Wang et al., 2023a).

In the current era of foundation models, a common strategy to solve machine learning problems is to adopt a pre-trained model and then fine-tune it based on specific downstream training data. Existing WSL approaches have paid most attention to the design of loss functions, using either unbiased risk estimators (UREs) with solid theoretical guarantees (Niu et al., 2016; Kiryo et al., 2017; Wang et al., 2024b) or effective regularization techniques to improve classification performance (Berthelot et al., 2019; Wang et al., 2022a; Li et al., 2022; Wang et al., 2023b). A common benchmark solution is to train a deep neural network from scratch to validate the effectiveness of the proposed loss function. However, the classification performance may degenerate significantly when using more complex datasets. This motivates us to consider a natural and practical research question: *Can we fine-tune pre-trained models with weakly supervised data to achieve better performance?*

In this paper, we attempt to investigate how pre-trained models can be appropriately exploited for WSL problems. First, we find that by using GPT-4o (OpenAI, 2024) to summarize and enrich the class description, and then directly applying a zero-shot Contrastive Language-Image Pre-Training (CLIP) model, we can achieve better classification performance than some state-of-the-art (SOTA) WSL algorithms trained from scratch (see Section 3.1). Such an observation indicates

that the performance of WSL approaches can be improved greatly by exploiting pre-trained models appropriately. To further improve performance on more complex datasets, it may often be beneficial to fine-tune pre-trained models with downstream weakly supervised training data. However, we find that naively fine-tuning pre-trained models with off-the-shelf unbiased or corrected risk estimators can often lead to inferior performance (see Section 3.2). We identify several causes of the shortcomings of existing risk estimators, including *overfitting exacerbation* and *feature degeneration*, which severely limit the effectiveness of model fine-tuning.

To this end, we propose a weakly supervised fine-tuning (WSFT) approach to effectively fine-tune pre-trained models for WSL problems (see Section 3.3). To overcome the shortcomings of existing risk estimators during the fine-tuning process, we directly perform empirical risk minimization by selecting training data with reliable labels. More specifically, we perform sample selection by distilling reliable supervision from the classifier obtained as a minimizer of the corrected risk estimator. Then, we alternately perform efficient model fine-tuning. The supervision distillation process and model fine-tuning are synergistically encapsulated in a unified framework and are mutually beneficial to improve the downstream classification performance. Our contributions can be summarized as follows:

- We find that pre-trained models can significantly outperform SOTA WSL approaches on benchmark datasets, suggesting a way to improve benchmark evaluation of WSL.
- We propose a WSFT approach to effectively fine-tune pre-trained models for various WSL problems. WSFT encapsulates reliable supervision distillation and efficient model fine-tuning seamlessly supported by theoretical guarantees.
- Extensive experiments on various WSL problems and benchmark datasets validate the effectiveness of our proposed method against SOTA WSL approaches (see Section 4).

## 2 PRELIMINARIES

In this section, we first describe the problem setting considered in this paper. Then, we introduce the background of binary classification and UU learning.

### 2.1 PROBLEM SETTING

From the perspective of the data generation process, WSL can be roughly categorized into two main formulations (Sugiyama et al., 2022), i.e., mutually contaminated distributions (MCD) (Scott et al., 2013; Katz-Samuels et al., 2019) and class-conditional noise (CCN) (Natarajan et al., 2013; 2018; Van Rooyen & Williamson, 2018). In particular, CCN can be shown to be a special case of MCD (Menon et al., 2015). Without loss of generality, we mainly discuss UU learning (Lu et al., 2019), the most general formulation of MCD in this paper. It has been proved that a wide variety of WSL problems are special cases of UU learning (Chiang & Sugiyama, 2023), including PU learning, pairwise-comparison (Pcomp) (Feng et al., 2021) learning, and similarity-unlabeled (SU) learning (Bao et al., 2018). Notably, our proposal can handle any kind of weak supervision generated based on MCD or CCN, and can be extended to *multi-class* setting.

### 2.2 BINARY CLASSIFICATION

Let $\mathcal{X} = \mathbb{R}^d$ denote the $d$-dimensional feature space and $\mathcal{Y} = \{+1, -1\}$ denote the label space. Let $p(\boldsymbol{x}, y)$ denote a joint probability density over the random variables $(\boldsymbol{x}, y) \in \mathcal{X} \times \mathcal{Y}$. Let $\pi_{\text{Te}} = p(y = +1)$ denote the class prior probability for the positive class. Besides, let $p_+(\boldsymbol{x}) = p(\boldsymbol{x}|y = +1)$ and $p_-(\boldsymbol{x}) = p(\boldsymbol{x}|y = -1)$ denote the class-conditional probability densities of positive and negative data, respectively. When using a deep model, our goal is to minimize the *ordinary classification risk*

$$R(\boldsymbol{\theta}, \boldsymbol{\omega}) = \mathbb{E}_{p(\boldsymbol{x}, y)} \left[ \ell \left( g \left( \boldsymbol{f} \left( \boldsymbol{x} \right) \right), y \right) \right]. \tag{1}$$

Here, $\mathbb{E}$ is the expectation operator, $\boldsymbol{f} : \mathbb{R}^d \rightarrow \mathbb{R}^m$ is an image encoder that outputs an $m$-dimensional feature vector, and $\boldsymbol{\theta} \in \Theta$ is learnable parameters, where $\Theta$ denotes the space of learnable parameters. Also, $g : \mathbb{R}^m \rightarrow \mathbb{R}$ is a classification head parameterized by $\boldsymbol{\omega} \in \Omega$, where $\Omega$ is the parameter space for $g$. Let $\sigma(\cdot)$ denote the sigmoid function, then $\sigma \left( g \left( \boldsymbol{f} \left( \boldsymbol{x} \right) \right) \right)$ denotes

an estimated probability of the example being positive. Also, $\ell$ is a *classification-calibrated* loss function (Bartlett et al., 2006), such as the cross-entropy loss.

## 2.3 UNLABELED-UNLABELED LEARNING

In UU learning, we are given two unlabeled training datasets $\mathcal{D}_1 = \{\mathbf{x}_{1,i}\}_{i=1}^{n_1}$ and $\mathcal{D}_2 = \{\mathbf{x}_{2,i}\}_{i=1}^{n_2}$ sampled from two mixture densities $p_1(\boldsymbol{x}) = \pi_1 p_+(\boldsymbol{x}) + (1-\pi_1)p_-(\boldsymbol{x})$ and $p_2(\boldsymbol{x}) = \pi_2 p_+(\boldsymbol{x}) + (1-\pi_2)p_-(\boldsymbol{x})$, with $\pi_1 \neq \pi_2$. Note that $\pi_1$ and $\pi_2$ can be estimated by off-the-shelf mixture proportion estimation methods (Ramaswamy et al., 2016; Garg et al., 2021; Cai et al., 2023). To train a binary classifier from $\mathcal{D}_1 \bigcup \mathcal{D}_2$, Lu et al. (2019) proposed an URE for UU learning, i.e.,

$$\widehat{R}_{\text{UU}}(\boldsymbol{\theta}, \boldsymbol{\omega}) = A\frac{1}{n_1}\sum_{i=1}^{n_1}\ell\left(g\left(\boldsymbol{f}\left(\boldsymbol{x}_{1,i}\right)\right), +1\right) - B\frac{1}{n_1}\sum_{i=1}^{n_1}\ell\left(g\left(\boldsymbol{f}\left(\boldsymbol{x}_{1,i}\right)\right), -1\right)$$

$$- E\frac{1}{n_2}\sum_{i=1}^{n_2}\ell\left(g\left(\boldsymbol{f}\left(\boldsymbol{x}_{2,i}\right)\right), +1\right) + F\frac{1}{n_2}\sum_{i=1}^{n_2}\ell\left(g\left(\boldsymbol{f}\left(\boldsymbol{x}_{2,i}\right)\right), -1\right), \quad (2)$$

where $A = (1-\pi_2)\pi_{\text{Te}}/(\pi_1-\pi_2)$, $B = \pi_2(1-\pi_{\text{Te}})/(\pi_1-\pi_2)$, $E = (1-\pi_1)\pi_{\text{Te}}/(\pi_1-\pi_2)$, and $F = \pi_1(1-\pi_{\text{Te}})/(\pi_1-\pi_2)$. Then $\widehat{R}_{\text{UU}}(\boldsymbol{\theta}, \boldsymbol{\omega})$ can be an URE of many WSL problems by using different values of the coefficients. Lu et al. (2020) found that when using deep models, overfitting problems would degrade the classification performance due to the negative terms in Eq. (2). Therefore, a corrected risk estimator was proposed to improve the classification performance:

$$\widehat{R}_{\text{CUU}}(\boldsymbol{\theta}, \boldsymbol{\omega}) = h\left(A\frac{1}{n_1}\sum_{i=1}^{n_1}\ell\left(g\left(\boldsymbol{f}\left(\boldsymbol{x}_{1,i}\right)\right), +1\right) - E\frac{1}{n_2}\sum_{i=1}^{n_2}\ell\left(g\left(\boldsymbol{f}\left(\boldsymbol{x}_{2,i}\right)\right), +1\right)\right)$$

$$+ h\left(F\frac{1}{n_2}\sum_{i=1}^{n_2}\ell\left(g\left(\boldsymbol{f}\left(\boldsymbol{x}_{2,i}\right)\right), -1\right) - B\frac{1}{n_1}\sum_{i=1}^{n_1}\ell\left(g\left(\boldsymbol{f}\left(\boldsymbol{x}_{1,i}\right)\right), -1\right)\right), \quad (3)$$

where $h$ is a non-negative risk-correction function, such as the ReLU function or the absolute value function (Lu et al., 2020; Wang et al., 2023a).

## 3 METHODOLOGY

In this section, we first introduce a zero-shot baseline based on the CLIP model, which can outperform SOTA approaches in the WSL literature. Then, we discuss that fine-tuning with existing risk estimators may not be optimal. Finally, we present our proposed WSFT approach.

### 3.1 A ZERO-SHOT BASELINE

Current WSL algorithms train deep neural networks from scratch, such as convolutional neural networks (CNNs) (Li et al., 2021b) and ResNet (He et al., 2016). SOTA algorithms can sometimes achieve comparable performance to supervised learning on these datasets. However, their performance may still be unsatisfactory on more complex and challenging datasets. In the era of foundation models, it has not been clear whether performance can be improved by exploiting the rich knowledge of pre-trained models, which are easily accessible and can be used without any fine-tuning since the class embeddings are readily available. Therefore, we conducted a preliminary pilot study using a zero-shot baseline approach without any fine-tuning.

Since existing benchmarks for binary classification mainly synthesize experimental datasets by partitioning multi-class data into binary classes, the semantic information of binary classes may be versatile and decentralized (Krizhevsky & Hinton, 2009; Helber et al., 2019). Therefore, inspired by Menon & Vondrick (2023) and Pratt et al. (2023), we asked GPT-4o to summarize and enrich the class description of positive and negative classes, and obtained text prompts as $t_k = $ `A photo of <class_description>` with $k \in \mathcal{Y}$. Then, we directly used CLIP for zero-shot classification. The algorithmic details can be found in Appendix A. Specifically, for a test image $\boldsymbol{x}_*$, we obtained its image embedding $\boldsymbol{f}(\boldsymbol{x}_*)$ using the image encoder of CLIP, and text embeddings $\boldsymbol{\psi}(t_k)$ $(k \in \mathcal{Y})$ of positive and negative classes using the text encoder of CLIP. Then,

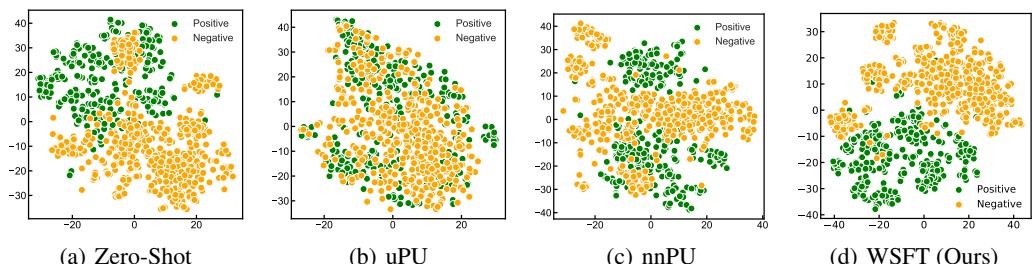

(a) Zero-Shot      (b) uPU      (c) nnPU      (d) WSFT (Ours)

Figure 2: The t-SNE visualizations of feature representations using the zero-shot baseline or different fine-tuning approaches on the PU learning setting of CIFAR-100.

we returned the prediction $y_*$ as the class whose embedding has the largest cosine similarity to the image embedding of $\boldsymbol{x}_*$, i.e.,

$$y_* = \arg\max_{k \in \mathcal{Y}} \cos\left(\boldsymbol{f}\left(\boldsymbol{x}_*\right), \boldsymbol{\psi}\left(t_k\right)\right). \tag{4}$$

Figure 1 shows the performance comparison between the zero-shot baseline and the SOTA approach GLWS on five WSL tasks (Chen et al., 2024) on CIFAR-10 and CIFAR-100, respectively. We found that the zero-shot baseline can already achieve better performance than current SOTA approaches by a large margin. Such an observation indicates that the use of pre-trained models can greatly improve the upper bounds of model performance for many WSL problems, since fine-tuning pre-trained models can often lead to better downstream performance than zero-shot models without fine-tuning (Menghini et al., 2023).

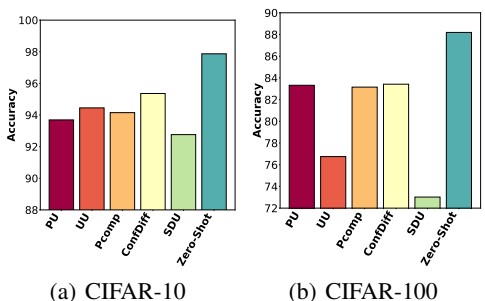

(a) CIFAR-10      (b) CIFAR-100

Figure 1: Comparison between the SOTA approach GLWS for each WSL problem (Chen et al., 2024) and the zero-shot baseline.

### 3.2 FINE-TUNING WITH EXISTING RISK ESTIMATORS IS SUBOPTIMAL

UREs, e.g., Eq. (2) and uPU (du Plessis et al., 2015), are mainstream solutions for WSL, where their universality and effectiveness have been demonstrated in a wide range of WSL problems (Sugiyama et al., 2022). Corrected risk estimators (CREs), e.g., Eq. (3) and nnPU (Kiryo et al., 2017), which wrap potentially negative terms with non-negative risk correction functions, can further improve the classification performance in many cases (Lu et al., 2020; Wang et al., 2023a). Therefore, it is a natural idea to directly apply off-the-shelf UREs and CREs for fine-tuning. However, we have found that such straightforward tuning strategies cannot lead to satisfactory results.

Here, we take PU learning as an example. Figure 3 shows positive and negative recall curves of unlabeled training data and the test curve of fine-tuning with UREs and CREs of PU learning on CIFAR-100, respectively. We can see that both risk estimators encounter *overfitting exacerbation* problems. First, uPU with CLIP as the backbone encounters more severe overfitting problems than commonly used network architectures in the literature (Chen et al., 2024). This indicates that UREs may not be suitable for fine-tuning large pre-trained models. Second, although nnPU can reduce the negative influence of overfitting, it still

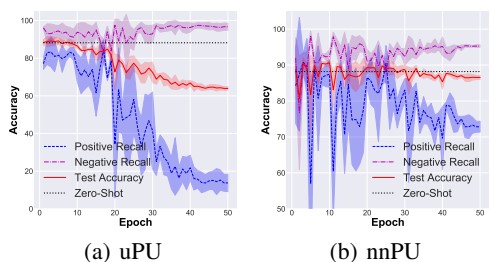

(a) uPU      (b) nnPU

Figure 3: Test curves and positive and negative recall curves for uPU and nnPU on the CIFAR-100 dataset.

shows inferior performance to the zero-shot baseline. Third, we can see that the overfitting problem is mainly due to the overfitting problems of positive classes. Figure 2 shows the t-SNE visualizations of feature representations learned with different strategies, including the zero-shot baseline as well as different fine-tuning approaches for the PU learning setting of CIFAR-100. We can see that both

risk estimators also encounter *feature degeneration* problems. We can observe that the zero-shot baseline initially learns a rather good clustering result. However, after using UREs and CREs for fine-tuning, representations of the two classes are not clearly separated and the clustering results are worse than those without fine-tuning.

Both overfitting exacerbation and feature degeneration suggest that fine-tuning with existing UREs and CREs may not be optimal. We suspect that the reason for both problems is due to the *mismatched loss terms* in the original UREs, which consider unlabeled examples as the wrong classes. For example, consider PU learning by setting $\pi_1 = 1$ and $\pi_2 = \pi_{\text{Te}}$. Then the fourth term in Eq. (2) will consider positive data from $\mathcal{D}_2$ as negative, which may cause overfitting problems of the positive class due to the *memorization effect* (Han et al., 2018; Zhang et al., 2021b) of deep neural networks. Although the disadvantage is not significant for moderate-sized models, it may be more obvious for very deep models.

### 3.3 A WEAKLY SUPERVISED FINE-TUNING APPROACH

To overcome the deficiency of UREs and CREs, a natural remedy is to perform ordinary *empirical risk minimization* directly on labeled data, since it does not involve mismatched loss terms. In this way, each class can learn discriminative feature representations, which will contribute to better performance. However, since true labels may not be accessible in WSL, it is necessary to investigate how to select reliable labeled data. In this paper, we rely on high-confidence predictions made by off-the-shelf classifiers, which have been shown to be a type of reliable supervision information widely adopted in the literature (Chen et al., 2020b; Zhang et al., 2021a; Kim et al., 2022).

Our approach consists of two alternating updating steps, i.e., *supervision distillation* and *model fine-tuning*. First, a classification head $g_1$ parameterized by $\boldsymbol{\omega}_1$ is updated by minimizing the CRE $\widehat{R}_1(\boldsymbol{\theta}, \boldsymbol{\omega}_1)$, which is instantiated specifically for different WSL problems, e.g., $\widehat{R}_{\text{CUU}}$ for UU learning in Eq. (3). However, instead of using it directly for test prediction, we use it only for eliciting reliable supervision to guide the subsequent classifier training. Specifically, we consider high-confidence predictions made by $g_1$ on the weakly supervised dataset $\widehat{D}_{\text{W}}$ as refined supervision. Here, $\widehat{D}_{\text{W}}$ could contain all the weakly supervised data whose true labels are not accessible to the learning algorithm, e.g., $\widehat{D}_{\text{W}} = \mathcal{D}_1 \bigcup \mathcal{D}_2$ for UU learning. We introduce $\widehat{D}_+ = \left\{\boldsymbol{x}_i | \boldsymbol{x}_i \in \widehat{D}_{\text{W}}, \sigma(g_1(\boldsymbol{f}(\boldsymbol{x}_i))) > \tau\right\}$ as the set of high-confidence positive data, and $\widehat{D}_- = \left\{\boldsymbol{x}_j | \boldsymbol{x}_j \in \widehat{D}_{\text{W}}, \sigma(g_1(\boldsymbol{f}(\boldsymbol{x}_j))) < 1 - \tau\right\}$ as the set of high-confidence negative data. Here, $\tau$ is set as a large threshold to filter out examples that may be mislabeled, which is an effective strategy in semi-supervised learning (SSL) to obtain reliable supervision (Sohn et al., 2020; Wang et al., 2023c; Chen et al., 2023). We call this process *supervision distillation*, since our supervision information to guide the classifier training has been progressively distilled from another classifier trained based on CREs. Based on the distilled supervision information, we train another classification head $g_2$ parameterized by $\boldsymbol{\omega}_2$ by minimizing the risk estimator:

$$\widehat{R}_2(\boldsymbol{\theta}, \boldsymbol{\omega}_2) = \frac{\pi_{\text{Te}}}{\left|\widehat{D}_+\right|} \sum_{\boldsymbol{x}_i \in \widehat{D}_+} \ell(g_2(\boldsymbol{f}(\boldsymbol{x}_i)), +1) + \frac{1 - \pi_{\text{Te}}}{\left|\widehat{D}_-\right|} \sum_{\boldsymbol{x}_j \in \widehat{D}_-} \ell(g_2(\boldsymbol{f}(\boldsymbol{x}_j)), -1), \quad (5)$$

where $|\cdot|$ is the cardinality.

In summary, the overall training objective is

$$L(\boldsymbol{\theta}, \boldsymbol{\omega}_1, \boldsymbol{\omega}_2) = \widehat{R}_1(\boldsymbol{\theta}, \boldsymbol{\omega}_1) + \widehat{R}_2(\boldsymbol{\theta}, \boldsymbol{\omega}_2). \quad (6)$$

The algorithmic details can be found in Appendix A, and the overall pipeline of WSFT is shown in Figure 5. In particular, two classification heads are updated independently, while the image encoder is shared and fine-tuned by minimizing the loss functions of both heads. After each round of model fine-tuning, we construct $\widehat{D}_+$ and $\widehat{D}_-$ by exploiting the predictions of $g_1$ on the weakly supervised training set. Supervision distillation and model fine-tuning are performed iteratively and are mutually beneficial for achieving better classification performance.

**Discussion.** Different from self-training in the SSL and domain adaptation literature (Lee et al., 2013; Zou et al., 2019; Wei et al., 2021), our approach adopts two classification heads to separate

the generation of pseudo-labels from the training of the classifier. The advantage is that this can effectively mitigate the *confirmation bias* that widely exists when training the classifier and generating pseudo-labels with a single classifier (Arazo et al., 2020; Chen et al., 2022). Moreover, unlike some SSL approaches that try to mitigate the confirmation bias by constructing teacher models with different strategies (Laine & Aila, 2017; Tarvainen & Valpola, 2017; Xie et al., 2020), the classifier used to generate pseudo-labels is not explicitly exposed to pseudo-labels.

**Extension.** Although this paper discusses binary classification, without loss of generality, our approach can be easily extended to multi-class classification problems. In particular, for weakly supervised multi-class classification (Cour et al., 2011; Lu et al., 2022; Wang et al., 2024b), $\widehat{R}_1 (\boldsymbol{\theta}, \boldsymbol{\omega}_1)$ can be instantiated accordingly with the CREs for specific WSL problems. Also, the supervision distillation procedure should be adapted to the multi-class classification setting.

**Theoretical analysis.** Since the analysis of $\widehat{R}_1 (\boldsymbol{\theta}, \boldsymbol{\omega})$ has been investigated thoroughly (Lu et al., 2020; Sugiyama et al., 2022), we focus on the theoretical properties of the proposed risk estimator $\widehat{R}_2 (\boldsymbol{\theta}, \boldsymbol{\omega})$ by providing its consistency and the convergence rate. Let $R_2 (\boldsymbol{\theta}, \boldsymbol{\omega}) = \mathbb{E}_{\widehat{D}_+, \widehat{D}_-} \left[ \widehat{R}_2 (\boldsymbol{\theta}, \boldsymbol{\omega}) \right]$ denote the expected version of the risk estimator proposed in Eq. (5), and $R_{0-1}^*$ denote the Bayes error of the dataset. We also assume that there exists some constant $C_{\mathfrak{R}}$ such that the Rademacher complexity $\mathfrak{R}_n(\mathcal{F})$ satisfies $\mathfrak{R}_n(\mathcal{F}) \leqslant C_{\mathfrak{R}}/\sqrt{n}$ (Golowich et al., 2018) where $\mathcal{F}$ is an arbitrary class of functions. All the proofs of the theorems can be found in Appendix F.

**Theorem 1.** *Assume there exists a constant $C_g$ such that $\sup \|g\|_\infty \leqslant C_g$ and some a $C_\ell$ such that $\sup_{|z| \leqslant C_g} \ell(z, \cdot) \leqslant C_\ell$. Consider the case where the marginal density and label distribution of $\widehat{D}_{\mathrm{W}}$ are the same as those of the test data. Also, we consider that $\widehat{D}_{\mathrm{W}}$ is directly labeled by $g_1$, which minimizes the CRE. For any $\delta > 0$, there exists a function $\varphi(\cdot)$ such that the following inequality holds with probability $1 - 2\delta$:*

$$|R_2 (\boldsymbol{\theta}, \boldsymbol{\omega}) - R (\boldsymbol{\theta}, \boldsymbol{\omega})| \leqslant C_\ell \left( R_{0-1}^* + \varphi^{-1} \left( \mathcal{O}_p \left( 1/\sqrt{n} \right) \right) \right). \tag{7}$$

*Here, $n$ is the number of training data for $g_1$, and $\varphi : [0, 1] \to [0, +\infty)$ is a non-decreasing and invertible function such that for any sequence $(u_i) \in [0, 1]$, $\varphi(u_i) \to 0$ if and only if $u_i \to 0$.*

*Remark* 1. If the Bayes error of the dataset is small enough and the number of training data is large enough, Theorem 1 shows that the risk can converge to the ordinary classification risk. Furthermore, when the number of training data goes to infinity, the optimal classifier of our proposed risk estimator in Eq. (5) can converge to that of the ordinary classification risk.

Let $\left( \widehat{\boldsymbol{\theta}}_2, \widehat{\boldsymbol{\omega}}_2 \right) = \arg\min_{\boldsymbol{\theta} \in \Theta, \boldsymbol{\omega} \in \Omega} \widehat{R}_2 (\boldsymbol{\theta}, \boldsymbol{\omega})$ denote the optimal classifier of the empirical risk estimator. Let $(\boldsymbol{\theta}_2^*, \boldsymbol{\omega}_2^*) = \arg\min_{\boldsymbol{\theta} \in \Theta, \boldsymbol{\omega} \in \Omega} R_2 (\boldsymbol{\theta}, \boldsymbol{\omega})$ denote the optimal classifier of the expected risk. Then we have the following theorem.

**Theorem 2.** *For any $\delta > 0$, the following inequality holds with probability $1 - \delta$:*

$$R_2 \left( \widehat{\boldsymbol{\theta}}_2, \widehat{\boldsymbol{\omega}}_2 \right) - R_2 (\boldsymbol{\theta}_2^*, \boldsymbol{\omega}_2^*) \leqslant \mathcal{O}_p \left( 1/\sqrt{\left| \widehat{D}_+ \right|} + 1/\sqrt{\left| \widehat{D}_- \right|} \right). \tag{8}$$

*Remark* 2. Theorem 2 elucidates the estimation error bound of the proposed risk estimator. As the number of training data goes to infinity, the risk of the optimal classifier learned with the empirical risk estimator approaches that of the optimal classifier of the risk, and the convergence rate is the optimal parametric rate for empirical risk minimization without making additional assumptions (Mendelson, 2008).

## 4 EXPERIMENTS

In this section, we perform experiments to evaluate the effectiveness of WSFT on various WSL problems, including PU learning, Pcomp learning, and UU learning.

### 4.1 EXPERIMENTAL SETUP

**Datasets.** In this work, we have considered more complex benchmark datasets than existing WSL literature, including CIFAR100 (Krizhevsky & Hinton, 2009), EuroSAT (Helber et al., 2019),

Table 1: Classification accuracy (mean±std) of each method on benchmark datasets for PU learning, where the best performance (excluding the zero-shot baseline) is shown in bold.

| Dataset | CIFAR-100-a | | EuroSAT-a | | Oxford-IIIT Pet-a | |
|---|---|---|---|---|---|---|
| # P/U | 40/40000 | 80/40000 | 30/9000 | 60/9000 | 95/2220 | 190/2220 |
| uPU (du Plessis et al., 2015) | 40.00±0.00 | 40.00±0.00 | 29.63±0.00 | 29.63±0.00 | 51.30±0.00 | 51.30±0.00 |
| nnPU (Kiryo et al., 2017) | 85.34±1.36 | 86.16±1.77 | 85.50±1.86 | 89.27±1.72 | 86.00±1.76 | 87.40±0.47 |
| VarPU (Chen et al., 2020a) | 60.00±0.00 | 60.00±0.00 | 70.40±0.00 | 70.40±0.00 | 48.70±0.00 | 48.70±0.00 |
| CVIR (Garg et al., 2021) | 78.52±1.03 | 88.10±1.40 | 91.37±0.15 | 92.47±0.15 | 88.48±0.77 | 90.67±1.28 |
| DistPU (Zhao et al., 2022b) | 78.73±1.48 | 80.96±3.05 | 47.97±3.21 | 51.88±2.00 | 92.46±0.14 | 92.71±0.58 |
| Count Loss (Shukla et al., 2023) | 79.70±1.19 | 84.71±2.32 | 87.36±2.29 | 90.00±1.16 | 84.93±0.73 | 87.85±0.71 |
| GLWS (Chen et al., 2024) | 79.71±0.55 | 85.20±1.09 | 84.38±2.11 | 91.58±0.42 | 88.87±1.66 | 91.94±0.85 |
| Zero-Shot | 88.19±0.00 | 88.19±0.00 | 73.01±0.00 | 73.01±0.00 | 90.11±0.00 | 90.11±0.00 |
| WSFT | **94.29±0.71** | **92.96±2.95** | **91.57±0.98** | **94.68±1.22** | **93.03±1.75** | **93.62±1.20** |
| Dataset | CIFAR-100-b | | EuroSAT-b | | Oxford-IIIT Pet-b | |
| # P/U | 60/40000 | 120/40000 | 70/9000 | 140/9000 | 90/2220 | 180/2220 |
| uPU (du Plessis et al., 2015) | 60.00±0.00 | 60.00±0.00 | 70.37±0.00 | 70.37±0.00 | 48.68±0.00 | 48.68±0.00 |
| nnPU (Kiryo et al., 2017) | 85.97±1.17 | 85.75±3.16 | 84.35±3.61 | 87.27±1.12 | 86.71±0.40 | 87.23±1.24 |
| VarPU (Chen et al., 2020a) | 40.00±0.00 | 40.00±0.00 | 29.60±0.00 | 29.60±0.00 | 51.30±0.00 | 51.30±0.00 |
| CVIR (Garg et al., 2021) | 82.38±0.33 | 85.06±2.45 | 86.29±2.88 | 92.41±1.23 | 91.06±1.04 | **94.27±0.60** |
| DistPU (Zhao et al., 2022b) | 65.75±8.13 | 73.84±12.07 | 70.37±0.00 | 70.37±0.00 | 91.54±0.53 | 92.88±1.31 |
| Count Loss (Shukla et al., 2023) | 83.17±2.52 | 86.35±1.93 | 82.60±4.02 | 86.03±1.61 | 83.50±0.80 | 88.17±3.18 |
| GLWS (Chen et al., 2024) | 84.84±1.14 | 88.03±1.32 | 86.66±2.58 | 89.66±1.87 | 87.14±0.30 | 89.82±0.83 |
| Zero-Shot | 88.19±0.00 | 88.19±0.00 | 73.01±0.00 | 73.01±0.00 | 90.11±0.00 | 90.11±0.00 |
| WSFT | **93.09±2.39** | **93.51±0.08** | **91.22±2.54** | **96.63±0.68** | **91.65±0.61** | 92.43±0.91 |

Oxford-IIIT Pet (Parkhi et al., 2012), and Caltech-101 (Fei-Fei et al., 2004). Details of these datasets can be found in Appendix B.1. Since these datasets are multi-class classification datasets, we transformed them into binary-class datasets. We used GPT-4o (OpenAI, 2024) to split the classes in each dataset into two groups. Detailed prompts and class splitting for each dataset on each setting can be found in Appendix B.2.

**Implementation Details.** We used the vision encoder of CLIP ViT B/16 (Dosovitskiy et al., 2020) as the backbone. We used visual prompt tuning (Jia et al., 2022), a representation parameter-efficient fine-tuning method as the fine-tuning method, and the number of learnable prompts was set to 10. Since it is non-trivial to perform model selection in WSL, we used the same parameters for all experiments. We used the cross-entropy loss coupled with label smoothing (LS) (Müller et al., 2019) as the loss function and RandAugment (Cubuk et al., 2020) to further improve the performance. Ablation studies and experiments with other pre-trained models can be found in Section 4.5. We re-implemented all the methods with the pre-trained model for fair comparisons. We set the learning rate to 0.1, the batch size to 64, and the threshold $\tau$ to 0.9 for all experiments. We used stochastic gradient descent as the optimizer and used the same optimizer and batch size for other methods. All experiments were run three times and we reported the average accuracy and standard deviation. More implementation details can be found in Appendix D.

## 4.2 POSITIVE-UNLABELED LEARNING

In this subsection, we provide experimental results on a PU learning problem. We considered the two-sample setting of PU learning (Niu et al., 2016), where a small number of positive examples were randomly selected as the positive training set and a large number of unlabeled data were separately sampled as the unlabeled training set. For compared methods, we chose SOTA PU learning approaches, including uPU (du Plessis et al., 2015), nnPU (Kiryo et al., 2017), VarPU (Chen et al., 2020a), CVIR (Garg et al., 2021), DistPU (Zhao et al., 2022b), Count Loss (Shukla et al., 2023), GLWS (Chen et al., 2024), and the zero-shot baseline.

The experimental results on PU learning are shown in Table 1. We can observe that WSFT achieves the best performance in 9 out of 10 settings. For example, compared with the best compared method GLWS (Chen et al., 2024), WSFT brings an 8.59% improvement on CIFAR-100-a with 40 positive training data. These results demonstrate that the proposed WSFT approach is more effective for fine-tuning pre-trained models under the PU learning setting.

Table 2: Classification accuracy (mean±std) of each method for Pcomp learning on the Caltech dataset (upper) and the CIFAR-100 dataset (lower), where the best performance (excluding the zero-shot baseline) is shown in bold.

| Dataset | Caltech-101 | | | | | |
|---|---|---|---|---|---|---|
| # U Pairs | 500 | | 1000 | | 1500 | |
| Class Priors | 0.4 | 0.5 | 0.4 | 0.5 | 0.4 | 0.5 |
| Pcomp-Unbiased (Feng et al., 2021) | 36.88±0.00 | 50.00±0.00 | 36.88±0.00 | 50.00±0.00 | 48.87±20.77 | 56.91±11.97 |
| Pcomp-Relu (Feng et al., 2021) | 78.00±2.79 | 72.70±2.89 | 75.84±2.06 | 73.82±1.10 | 76.60±1.12 | 74.33±1.58 |
| Pcomp-ABS (Feng et al., 2021) | 71.41±2.95 | 65.60±0.70 | 77.66±3.03 | 74.02±1.11 | 74.33±1.58 | 76.28±3.15 |
| GLWS (Chen et al., 2024) | 71.47±2.53 | 72.54±1.49 | 77.38±1.10 | 77.20±0.69 | 82.13±2.10 | 81.08±1.16 |
| Zero-shot | 83.57±0.00 | 86.74±0.00 | 83.57±0.00 | 86.74±0.00 | 83.57±0.00 | 86.74±0.00 |
| WSFT | **78.57±2.56** | **80.80±5.78** | **86.08±3.27** | **88.63±0.56** | **91.49±1.68** | **94.87±0.99** |
| Dataset | CIFAR-100 | | | | | |
| # U Pairs | 1000 | | 5000 | | 10000 | |
| Class Priors | 0.4 | 0.5 | 0.4 | 0.5 | 0.4 | 0.5 |
| Pcomp-Unbiased (Feng et al., 2021) | 46.67±11.55 | 62.36±21.41 | 53.90±12.04 | 71.98±19.03 | 84.28±1.57 | 85.64±1.69 |
| Pcomp-Relu (Feng et al., 2021) | 73.44±2.71 | 72.85±1.78 | 78.62±2.90 | 77.68±1.19 | 88.21±1.07 | 87.30±1.27 |
| Pcomp-ABS (Feng et al., 2021) | 73.73±0.98 | 70.32±2.45 | 89.57±1.84 | 88.59±1.34 | 93.23±1.05 | 92.82±0.99 |
| GLWS (Chen et al., 2024) | 72.03±6.31 | 73.01±4.16 | 82.19±6.30 | 86.82±1.63 | 86.44±1.54 | 88.64±3.24 |
| Zero-shot | 88.19±0.00 | 87.04±0.00 | 88.19±0.00 | 87.04±0.00 | 88.19±0.00 | 87.04±0.00 |
| WSFT | **82.89±1.90** | **84.41±1.18** | **95.60±0.19** | **95.76±0.04** | **96.13±0.43** | **96.00±0.10** |

## 4.3 PAIRWISE-COMPARISON LEARNING

In this subsection, we consider the Pcomp problem framework. For Pcomp learning (Feng et al., 2021), we have unlabeled data pairs from (+1,+1), (+1,-1), and (-1,-1). Following previous experimental protocols (Feng et al., 2021), we set the class prior in advance and then sampled the data to generate training pairs accordingly. We used Pcomp with its variants (Feng et al., 2021), GLWS (Chen et al., 2024), and the zero-shot baseline as compared methods.

The experimental results are presented in Table 2. The proposed WSFT approach shows consistently superior performance compared with other approaches across all settings. For example, on the Caltech dataset with a class prior of 0.4, WSFT yields an accuracy improvement of 9.36% over the previous best performance using 1500 pairs. This clearly validates the effectiveness of the proposed method in Pcomp learning.

## 4.4 UNLABELED-UNLABELED LEARNING

In this subsection, we consider the UU learning setting for binary classification. We selected a certain number of unlabeled examples with two different class priors. For compared methods, we considered BER (Menon et al., 2015), UU-Unbiased (Lu et al., 2019), UU-ABS (Lu et al., 2020), GLWS (Chen et al., 2024), and the zero-shot baseline.

The experimental results are shown in Table 3. Compared with the other methods, WSFT shows overall consistent improvements across all settings. It is worth noting that WSFT can significantly outperform other methods when the number of examples is small. For example, WSFT yields an accuracy improvement of 6.98% on the CIFAR-100 dataset with 100 examples (class priors 0.8/0.2). This shows that WSFT can be effective for learning from a very small number of unlabeled data.

## 4.5 FURTHER ANALYSIS

**Other pre-trained models.** We have conducted experiments on other pre-trained models to test the generalization ability of the proposed WSFT. We used ViT pre-trained on ImageNet (Deng et al., 2009) as a backbone. We conducted experiments on the PU learning setting and compared our approach with two strong baselines CVIR (Garg et al., 2021) and GLWS (Chen et al., 2024). The results are shown in Table 4. WSFT still outperforms the other two methods on all three datasets. These results show that WSFT can be effectively combined with different pre-trained models.

Table 3: Classification accuracy (mean±std) of each method for UU learning on the CIFAR-100 dataset (upper) and the EuroSAT (lower) dataset, where the best performance (excluding the zero-shot baseline) is shown in bold.

| Dataset | CIFAR-100 | | | | | |
|---|---|---|---|---|---|---|
| # U | 100 | | 1000 | | 5000 | |
| Class Priors | 0.8/0.2 | 0.9/0.3 | 0.8/0.2 | 0.9/0.3 | 0.8/0.2 | 0.9/0.3 |
| BER (Menon et al., 2015) | 79.36±2.78 | 80.16±1.63 | 84.20±2.16 | 86.13±0.17 | 95.48±0.17 | 95.64±0.50 |
| UU-Unbiased (Lu et al., 2019) | 78.54±3.52 | 74.89±4.13 | 87.23±1.18 | 86.20±0.85 | 76.18±31.33 | 56.35±28.32 |
| UU-ABS (Lu et al., 2020) | 78.58±1.69 | 71.69±5.47 | 88.97±1.50 | 89.72±0.89 | 95.67±0.26 | 95.57±0.68 |
| GLWS (Chen et al., 2024) | 82.14±1.87 | 75.88±0.14 | 92.81±1.07 | 93.34±0.42 | 95.93±0.12 | 95.59±0.38 |
| Zero-shot | 88.19±0.00 | 88.19±0.00 | 88.19±0.00 | 88.19±0.00 | 88.19±0.00 | 88.19±0.00 |
| WSFT | **88.12±2.17** | **82.27±2.15** | **95.84±0.20** | **94.51±0.71** | **96.24±0.19** | **96.14±0.21** |
| Dataset | EuroSAT | | | | | |
| # U | 50 | | 100 | | 150 | |
| Class Priors | 0.8/0.2 | 0.9/0.3 | 0.8/0.2 | 0.9/0.3 | 0.8/0.2 | 0.9/0.3 |
| BER (Menon et al., 2015) | 73.61±1.18 | 77.39±1.85 | 76.48±3.54 | 78.84±1.20 | 77.20±2.77 | 83.60±1.62 |
| UU-Unbiased (Lu et al., 2019) | 79.01±4.54 | 29.63±0.00 | 81.07±2.02 | 29.63±0.00 | 64.42±30.50 | 29.63±0.00 |
| UU-ABS (Lu et al., 2020) | 81.74±3.83 | 77.22±2.63 | 82.94±3.46 | 78.36±1.84 | 85.53±2.41 | 84.28±2.40 |
| GLWS (Chen et al., 2024) | 85.00±3.18 | 77.73±11.56 | 90.09±2.34 | 76.29±9.92 | 92.86±2.38 | 88.56±2.32 |
| Zero-shot | 73.01±0.00 | 73.01±0.00 | 73.01±0.00 | 73.01±0.00 | 73.01±0.00 | 73.01±0.00 |
| WSFT | **89.03±5.55** | **84.15±2.97** | **94.73±0.93** | **92.61±1.58** | **97.05±0.38** | **96.16±1.37** |

Table 4: Classification accuracy (mean±std) of each method on benchmark datasets with Vit pre-trained on ImageNet as the backbone, where the best performance is shown in bold.

| Dataset | CIFAR-100-a | | EuroSAT-a | | Oxford-IIIT Pet-a | |
|---|---|---|---|---|---|---|
| # P/U | 40/40000 | 80/40000 | 30/9000 | 60/9000 | 95/2220 | 190/2220 |
| CVIR (Garg et al., 2021) | 73.60±0.65 | 81.36±1.76 | 83.53±2.53 | 91.67±0.46 | 81.60±1.71 | 88.55±2.65 |
| GLWS (Chen et al., 2024) | 80.19±1.63 | 83.39±1.10 | 80.13±0.93 | 88.79±0.57 | 89.74±1.11 | 92.45±0.61 |
| WSFT | **87.62±1.86** | **86.31±2.61** | **92.44±1.61** | **94.03±0.59** | **94.69±1.74** | **94.26±0.92** |

**Ablation study.** We examined the effectiveness of dual classification heads by using only a single classification head. We also examined the effectiveness of LS and RandAugment. We also tried two different inference strategies: using $g_1$ alone for inference ($g_1$ test) and using the ensemble of $g_1$ and $g_2$ for inference. The experimental results are given in Table 5. We can see that using two heads improves the accuracy by 3.82% on CIFAR-100-a with 90 positive samples and by 2.34% on EuroSAT-b with 70 positive examples. Moreover, even without using LS and RandAugment, our approach can still outperform the compared methods. Also, the use of the ensemble model can sometimes additionally improve the model performance.

**Hyperparameter sensitivity.** We examined the sensitivity of the hyperparameters for WSFT. Here, $\tau$ is the sample selection threshold and $\alpha$ is the label smoothing coefficient. The experimental results are shown in Figure 4. We can see that the performance is not sensitively affected by changes of $\tau$ and $\alpha$ within certain ranges.

## 5 RELATED WORK

In this section, we discuss related work on WSL, pre-trained models, applications of pre-trained models to WSL, and knowledge distillation.

**Weakly supervised learning.** In WSL, we want to train a model that can perform comparably to supervised learning under incomplete, inexact, or inaccurate supervision (Zhou, 2018; Sugiyama et al., 2022). WSL in machine learning is fundamentally different from weakly supervised segmentation (WSS) in computer vision (Ahn & Kwak, 2018; Lin et al., 2023), where WSS refers to segmenting images with image-level labels. In recent years, many WSL problems have been studied (Ratner et al., 2016; Lu et al., 2019; Bao et al., 2018; Feng et al., 2021; Zhang et al., 2021c; Li et al., 2023). The most common strategy for WSL is to rewrite the classification risk and then

Table 5: Experimental results of ablation studies. The class priors for the UU setting are 0.8/0.2, and the numbers of training data are 80/40000, 70/9000, 5000, and 150 respectively.

|  | PU | | UU | |
|---|---|---|---|---|
| Dataset | CIFAR-100-a | EuroSAT-b | CIFAR-100 | EuroSAT |
| WSFT | 95.12 | 92.72 | 96.11 | 97.11 |
| Single head | 91.30 | 90.48 | 94.35 | 96.10 |
| w/o LS | 91.81 | 90.71 | 96.80 | 95.71 |
| w/o RandAugment | 93.14 | 90.31 | 96.18 | 95.39 |
| $g_1$ test | 91.73 | 90.01 | 96.17 | 94.20 |
| Ensemble | 93.09 | 91.08 | 96.18 | 97.11 |

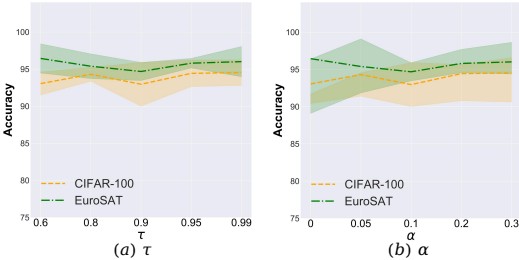

Figure 4: Hyperparameter sensitivity analysis w.r.t. $\alpha$ and $\tau$ on CIFAR-100 and EuroSAT.

derive an URE. It has typically been shown that the URE is a consistent risk estimator with respect to the expected classification risk and enjoys optimal convergence rates (Sugiyama et al., 2022). Although UREs are elegant and theoretically sound, a common drawback is that they often encounter overfitting problems. Another group of WSL methods aims to improve classification performance by designing various regularization techniques (Yu et al., 2021; Wu et al., 2024), such as contrastive learning (Li et al., 2021a; Wang et al., 2022a; Acharya et al., 2022). However, such complicated training strategies may not be computationally efficient when fine-tuning large pre-trained models.

**Pre-trained models.** Pre-trained models have achieved great success in many machine learning applications (Brown et al., 2020; Caron et al., 2021; Oquab et al., 2024; Jia et al., 2021; Bommasani et al., 2021). For example, the CLIP model (Radford et al., 2021) uses a contrastive loss to align image and text features within a common manifold, ensuring that semantically similar images and text are positioned close together. As a result, CLIP achieves high zero-shot classification accuracy, often comparable to or even better than supervised models (Radford et al., 2021). To further improve the performance of pre-trained models, various fine-tuning methods have been proposed (Jia et al., 2022; Zhou et al., 2022; Hu et al., 2022). While pre-trained models have achieved significant success in the standard supervised learning paradigm, their impact on WSL remains underexplored.

**Weakly supervised learning with pre-trained models.** Hendrycks et al. (2019) found that using pre-trained models can improve the model robustness to noisy labels. Yu et al. (2021) investigated how to fine-tune pre-trained language models for text classification tasks with only noisy and unlabeled data. However, their scope is limited to the classification problem of noisy and unlabeled text, and their training strategies are less efficient. Some recent works in SSL or noisy-label learning have tried to use or fine-tune pre-trained CLIP models and achieved better empirical performance (Wang et al., 2022b; Gan & Wei, 2024; Wang et al., 2024a; Ahn et al., 2024; Feng et al., 2024). However, their approaches can only be applied within a single problem setting and cannot be extended to solve other WSL problems.

**Knowledge distillation.** The goal of knowledge distillation (KD) (Hinton et al., 2015) is to train a student model by distilling knowledge from a teacher model (Zhang et al., 2018; Zhao et al., 2022a). Recently, KD has been applied to many areas, such as vision-language models to maintain the generalization ability of models (Yao et al., 2023; Li et al., 2024), SSL to leverage the teacher's knowledge (Sohn et al., 2020; Yang et al., 2024). In particular, the supervision distillation process in the proposed WSFT approach can be considered as a kind of knowledge distillation. However, the teacher model in WSFT is another classification head of the same size that is efficient and effective in mitigating confirmation bias.

## 6 CONCLUSION

In this paper, we investigated the impact of pre-trained models on WSL. Our empirical results showed that the use of pre-trained models, such as CLIP, can significantly improve the classification performance for WSL even without fine-tuning. To further improve the classification performance of pre-trained models, we proposed a novel WSFT approach that uses dual classification heads to alternately perform efficient model fine-tuning and supervision distillation. Extensive experiments on different WSL settings demonstrated the superiority of WSFT over SOTA WSL methods. A limitation of our approach is that it is specifically designed for image data. In the future, it is also promising to investigate how LLMs, another large category of pre-trained foundation models, affect and improve WSL.

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

Table 6: Dataset statistics and hand-crafted prompts.

| Dataset | Classes | Train | Val | Test | Hand-crafted prompt |
|---|---|---|---|---|---|
| CIFAR-100 | 100 | 50,000 | N/A | 10,000 | "a photo of a [CLASS]." |
| EuroSAT | 10 | 13,500 | 5,400 | 8,100 | "a centered satellite photo of [CLASS]." |
| Oxford-IIIT Pet | 37 | 2,944 | 736 | 3,669 | "a photo of a [CLASS], a type of pet." |
| Caltech-101 | 100 | 4,128 | 1,649 | 2,465 | "a photo of a [CLASS]." |

## A  ALGORITHMIC DETAILS

---
**Algorithm 1** Zero-Shot Baseline
---
**Input:** GPT-4o, Image encoder $\boldsymbol{f}$ of CLIP, text encoder $\boldsymbol{\psi}$ of CLIP, unseen instance $\boldsymbol{x}_*$, label space $\mathcal{Y}$.

1: **Obtain** the image embedding $\boldsymbol{f}(\boldsymbol{x}_*)$ of the unseen instance.
2: **for** $k \in \mathcal{Y}$ **do**
3:     **Generate** text descriptions $t_k$ using GPT-4o;
4:     **Obtain** the text embedding $\boldsymbol{\psi}(t_k)$ for class $k$;
5: **end for**
6: **Return** $y_* = \arg\max_{k \in \mathcal{Y}} \cos(\boldsymbol{f}(\boldsymbol{x}_*), \boldsymbol{\psi}(t_k))$;

**Output:** Predicted label $y_*$.

---
**Algorithm 2** Weakly Supervised Fine-Tuning
---
**Input:** Image encoder $\boldsymbol{f}$ of CLIP, classification head $\boldsymbol{g}_1$ and $\boldsymbol{g}_2$, unlabeled datasets $\mathcal{D}_1$ and $\mathcal{D}_2$, class priors $\pi_1$ and $\pi_2$, epoch $T_{\max}$, warm up epoch $T_{\mathrm{warm}}$, iteration $I_{\max}$.

1: **for** $t = 1, 2, \ldots, T_{\max}$ **do**
2:     **Shuffle** the unlabeled training datasets $\mathcal{D}_1$ and $\mathcal{D}_2$;
3:     **for** $j = 1, \ldots, I_{\max}$ **do**
4:         **Fetch** mini-batch $\mathcal{D}_{1,j}$ from $\mathcal{D}_1$ and $\mathcal{D}_{2,j}$ from $\mathcal{D}_2$;
5:         **if** $t \leqslant T_{\mathrm{warm}}$ **then**
6:             **Update** the classification head $g_1$ by minimizing $\widehat{R}_1(\boldsymbol{\theta}, \boldsymbol{\omega}_1)$;
7:         **else**
8:             **Update** the classification head $g_1$ and $g_2$ by minimizing Eq. (6);
9:             **Obtain** the high-confidence positive data $\widehat{D}_+$ and negative data $\widehat{D}_-$;
10:         **end if**
11:     **end for**
12: **end for**
13: **Return** Image encoder $\boldsymbol{f}$ of CLIP, Classification head $g_2$;

**Output:** Image encoder $\boldsymbol{f}$ of CLIP, Classification head $g_2$.

We provide the pseudo-code of zero-shot baseline in Algorithm 1, the pseudo-code of WSFT in Algorithm 2, and the overall pipeline of WSFT in Figure 5.

## B  DETAILS OF DATASETS

### B.1  DATASET STATISTICS

The statistics of the datasets used in experiments can be found in Table 6. For CIFAR-100, we use the original training and test dataset. For EuroSAT, Oxford-IIIT Pet-b, and Caltech-101, we use the training, validation, and test dataset splitting of CoOp (Zhou et al., 2022).

### B.2  DATASET SPLITTING

Since the datasets used in the experiment are multi-class, we transform them into binary classification datasets by dividing the classes into two groups: positive and negative. Given the large number

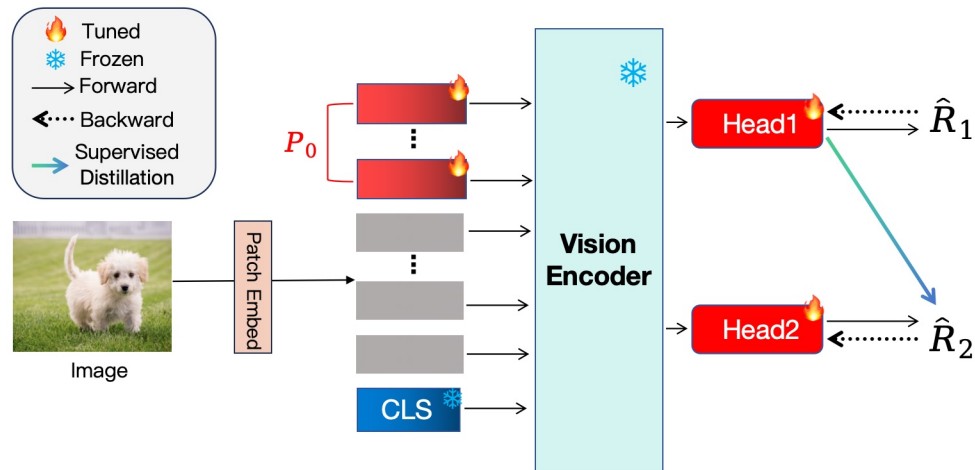

Figure 5: The overall pipeline of WSFT.

of classes, manual splitting is challenging, so we utilize GPT-4o to group the classes in each dataset. The grouping prompt is provided below:

grouping prompt: *I want to use the [DATASET_NAME] dataset for binary classification and need to divide the [CLASS_NUMBER] classes into two groups. The classes in each group should be semantically similar, while the classes between the two groups should be semantically different so that each new class is discriminative. The ratio of the amount of the data from these two new classes should be 4:6 or 5:5. Can you provide the detailed group information of this? Please provide a detailed step-by-step explanation, and you do not need to provide the python code.*

Specifically, for each dataset, we replace the dataset name with *[DATASET_NAME]* and the number of classes with *[CLASS_NUMBER]*.

Then, we provide the final Dataset splitting as below.

**CIFAR-100:**

**Group1 (40 classes):** [*trout, orchids, bowls, telephone, porcupine, oranges, cockroach, bear, oak, palm, aquarium fish, apples, clock, squirrel, train, roses, cans, butterfly, spider, dinosaur, pears, table, skyscraper, fox, boy, otter, poppies, bee, castle, tank, shark, sunflowers, road, elephant, bicycle, leopard, worm, mouse, maple, willow,* ]

**Group1 (60 classes):** [*beaver, dolphin, seal, whale, flatfish, ray, tulips, bottles, cups, plates, mushrooms, sweet peppers, computer keyboard, lamp, television, bed, chair, couch, wardrobe, beetle, caterpillar, lion, tiger, wolf, bridge, house, cloud, forest, mountain, plain, sea, camel, cattle, chimpanzee, kangaroo, possum, raccoon, skunk, crab, lobster, snail, baby, girl, man, woman, crocodile, lizard, snake, turtle, hamster, rabbit, shrew, pine, bus, motorcycle, pickup truck, lawn-mower, rocket, streetcar, tractor*]

**EuroSAT:**

**Group1 (3 classes):** [*Highway or Road, Industrial Buildings, Residential Buildings,* ]

**Group2 (7 classes):** [*Annual Crop Land, Forest, Herbaceous Vegetation Land, Pasture Land, Permanent Crop Land, River, Sea or Lake,* ]

**Oxford-IIIT Pet-b:**

**Group1 (19 classes):** [*abyssinian, bengal, birman, bombay, british_shorthair, chihuahua, egyptian_mau, havanese, japanese_chin, maine_coon, miniature_pinscher, persian, pomeranian, pug, ragdoll, russian_blue, siamese, sphynx, yorkshire_terrier* ]

**Group2 (18 classes):** [*american_bulldog, american_pit_bull_terrier, basset_hound, beagle, boxer, english_cocker_spaniel, english_setter, german_shorthaired, great_pyrenees, keeshond, leon-*

Table 7: Positive and negative label groups of datasets and the statistics of those PU training sets.

| Dataset | Positive Class Group | Negative Class Group | $\pi$ |
|---|---|---|---|
| CIFAR-100-a | Group-1 | Group-2 | 0.4 |
| CIFAR-100-b | Group-2 | Group-1 | 0.6 |
| EuroSAT-a | Group-1 | Group-2 | 0.3 |
| EuroSAT-b | Group-2 | Group-1 | 0.7 |
| Oxford-IIIT Pet-a | Group-1 | Group-2 | 0.51 |
| Oxford-IIIT Pet-b | Group-2 | Group-1 | 0.49 |

*berger, newfoundland, saint_bernard, samoyed, scottish_terrier, shiba_inu, staffordshire_bull_terrier, wheaten_terrier* ]

**Caltech-101**:

**Group1 (49 classes):** [*ant, bass, beaver, bonsai, brontosaurus, butterfly, cougar_body, cougar_face, crab, crayfish, crocodile, crocodile_head, dalmatian, dolphin, dragonfly, elephant, emu, flamingo, flamingo_head, garfield, gerenuk, hedgehog, hawksbill, ibis, joshua_tree, kangaroo, leopard, llama, lobster, lotus, mayfly, nautilus, octopus, okapi, panda, pigeon, platypus, rhino, rooster, scorpion, sea_horse, starfish, stegosaurus, strawberry, sunflower, tick, trilobite, water_lilly, wild_cat*]

**Group 2 (51 classes):** [accordion, airplane, anchor, barrel, binocular, brain, buddha, camera, cannon, car_side, ceiling_fan, cellphone, chair, chandelier, cup, dollar_bill, electric_guitar, euphonium, ewer, face, ferry, gramophone, grand_piano, headphone, helicopter, inline_skate, ketch, lamp, laptop, mandolin, menorah, metronome, minaret, motorbike, pagoda, pizza, pyramid, revolver, saxophone, schooner, scissors, snoopy, soccer_ball, stapler, stop_sign, umbrella, watch, wheelchair, windsor_chair, wrench, yin_yang]

**PU Learning Set-Up.** Dataset setup in PU learning is shown in Table 7.

## C  ZERO-SHOT PROMPT

In this section, we provide the prompt to generate text description for zero-shot CLIP binary classification. We ask GPT-4o to generate the text description, and the prompt is shown below.

prompt:  *I am doing a binary classification for [DATASET_NAME]. I divided the [DATASET_NUMBER] classes into two groups to consider them positive and negative. The positive classes are:  [POSITIVE_GROUP_NAME_LIST]. The negative classes are [NEGATIVE_GROUP_NAME_LIST]. Can you help me summarize each of the two classes so that I can use it for CLIP Zero-Shot Classification? The result should be two sentences, one for each class. Please provide rich and sufficient descriptions for each class, starting with ä photo of änd not exceeding the maximum token limit of CLIP. Please think step-by-step.*

Specifically, for each dataset, we replace the dataset name with *[DATASET_NAME]*, the number of classes with *[CLASS_NUMBER]*, the positive class name list with *[POSITIVE_GROUP_NAME_LIST]*, and the negative class name list with *[NEGATIVE_GROUP_NAME_LIST]*.

Finally, we provide the text prompts for positive and negative classes as below:

**CIFAR-100:**

**Positive:**  *various living organisms and plants, including aquatic mammals, fish, flowers, fruits, vegetables, insects, large carnivores, large omnivores and herbivores, medium-sized mammals, non-insect invertebrates, reptiles, small mammals, and trees.*

**Negative:**  *various man-made objects, scenes, and people, including vehicles, household electrical devices, household furniture, large man-made outdoor things, large natural outdoor scenes, people, small man-made outdoor things, and small man-made indoor things.*

**EuroSAT:**

Table 8: Results on ImageNet-100 on PU learning.

| Dataset | uPU | nnPU | VarPU | CVIR | DistPU | Count Loss | GLWS | Zero-Shot | WSFT |
|---|---|---|---|---|---|---|---|---|---|
| ImageNet-100-a | 40.00±0.00 | 85.43±1.95 | 60.00±0.00 | 92.60±1.87 | 88.27±1.45 | 79.93±1.95 | 87.97±2.46 | 75.26±0.00 | **92.90±2.07** |

Table 9: Results on CIFAR-100 on PU learning with a different partitioning strategy.

| Dataset | uPU | nnPU | VarPU | CVIR | DistPU | Count Loss | GLWS | Zero-Shot | WSFT |
|---|---|---|---|---|---|---|---|---|---|
| CIFAR-100-a | 56.00±0.00 | 80.73±4.51 | 44.00±0.00 | 74.07±2.95 | 70.97±13.00 | 77.83±3.91 | 78.17±1.01 | 72.29±0.00 | **87.64±1.91** |

**Positive:** *a centered satellite photo of man-made environments, featuring expansive highways, industrial complexes with large factories, and dense residential areas filled with houses and buildings, representing urban and developed landscapes.*

**Negative:** *a centered satellite photo of natural environments, including lush forests, sprawling pastures, agricultural lands like annual and permanent crops, vibrant herbaceous vegetation, flowing rivers, and serene lakes or seas, representing the harmony of diverse ecosystems.*

**Oxford-IIIT Pet**

**Positive:** *a photo of a small pet, including elegant cat breeds like Abyssinian, Bengal, Persian, Siamese, Sphynx, and toy dog breeds like Chihuahua, Pug, Pomeranian, Yorkshire Terrier, Havanese, known for their petite size and distinctive features, a type of pet.*

**Negative:** *a photo of a medium to large dog breed, such as energetic and robust breeds like American Bulldog, Beagle, Boxer, German Shorthaired Pointer, Saint Bernard, Samoyed, Shiba Inu, known for their strength and active nature, a type of pet.*

**Clatech-101**

**Positive:** *a photo of animals, plants, and other living things, including insects, mammals, birds, aquatic creatures, and plant species like bonsai and lotus.*

**Negative:** *a photo of man-made objects and vehicles, including items like airplanes, musical instruments, tools, cameras, laptops, and cars.*

## D    MORE IMPLEMENTATION DETAILS

In this section, we provide more implementation details. All the methods were implemented using Pytorch (Paszke et al., 2019). For a fair comparison, we fix the hyper-parameters when re-implementing previous methods. We set the learning rate as 0.001 for GLWS (Chen et al., 2024) on all experiments, following the official code. We set the learning rate as 0.1 for uPU (du Plessis et al., 2015),nnPU (Kiryo et al., 2017), VarPU (Chen et al., 2020a), Pcomp-Unbiased, Pcomp-Relu, Pcomp-ABS (Feng et al., 2021), BER (Menon et al., 2015), UU-Unbiased (Lu et al., 2019), and UU-ABS (Lu et al., 2020), 0.001 for CVIR (Garg et al., 2021), 0.0005 for DistPU (Zhao et al., 2022b), 0.00005 for Count Loss (Shukla et al., 2023). We follow FixMath (Sohn et al., 2020) for the use of strong augmentation for distillation. We use label smoothing with the parameter set as 0.1 for all experiments. All experiments were run on one single NVIDIA A100 GPU.

## E    MORE EXPERIMENTAL RESULTS

In this section, we provide more experimental results.

**Results on ImageNet-100.** We conducted experiments on ImageNet-100 in the PU learning setting. The number of labeled positive data is 120 and unlabeled data is 10000. The results are shown in Table 8. As shown in Table 8, the proposed WSFT outperforms all existing methods on the ImageNet-100 dataset.

**Results on CIFAR-100 with different partitioning.** We conducted experiments on CIFAR-100 in the PU learning setting with different group partitioning. The previous partitioning is "living organisms and plants" versus "man-made objects, scenes, and people". We asked GPT-4o to separate classes in CIFAR-100 with a different partitioning strategy. The partitioning is "Dynamic Entities

(Mobile or Potential for Motion)" versus "Static or Stationary Entities.". The results are shown in
Table 9. As shown in Table 9, the proposed WSFT still outperforms all existing methods on this
partition by a large margin.

# F   PROOFS

## F.1   PROOF OF THEOREM 1

Let $\left(\widehat{\boldsymbol{\theta}}_1, \widehat{\boldsymbol{\omega}}_1\right) = \arg\min_{\boldsymbol{\theta}\in\Theta, \boldsymbol{\omega}\in\Omega} \widehat{R}_1\left(\boldsymbol{\theta}, \boldsymbol{\omega}\right)$ denote the optimal classifier of the empirical risk estimator. Let $\left(\boldsymbol{\theta}^*, \boldsymbol{\omega}^*\right) = \arg\min_{\boldsymbol{\theta}\in\Theta, \boldsymbol{\omega}\in\Omega} R\left(\boldsymbol{\theta}, \boldsymbol{\omega}\right)$ denote the optimal classifier of the risk in Eq.(1).
We also assume that $\Omega$ is closed under negation. We also assume that the loss function $\ell(z, \cdot)$ is Lipschitz continuous w.r.t. $z$ with a Lipschitz constant $L_\ell$ and the risk correction function $h(z)$ is also
Lipschitz continuous w.r.t. $z$ with a Lipschitz constant $L_h$. We take UU learning as an example by
instantiating $\widehat{R}_1\left(\boldsymbol{\theta}, \boldsymbol{\omega}\right)$ with $\widehat{R}_{\mathrm{CUU}}\left(\boldsymbol{\theta}, \boldsymbol{\omega}\right)$. The estimation error bound of nnUU can be generalized
to a wide range of WSL problems (Chiang & Sugiyama, 2023).

**Lemma 1** (Restatement of Theorem 4 in Lu et al. (2020)). *For any $\delta > 0$, the following inequality
holds with probability $1 - \delta$:*

$$R\left(\widehat{\boldsymbol{\theta}}_1, \widehat{\boldsymbol{\omega}}_1\right) - R\left(\boldsymbol{\theta}^*, \boldsymbol{\omega}^*\right) \leqslant \mathcal{O}_p\left(1/\sqrt{n_1} + 1/\sqrt{n_2}\right). \tag{9}$$

The estimation error bounds of CREs for other WSL problems are similar (Kiryo et al., 2017; Lu
et al., 2020; Wang et al., 2023a). Without loss of generality, we assume

$$R\left(\widehat{\boldsymbol{\theta}}_1, \widehat{\boldsymbol{\omega}}_1\right) - R\left(\boldsymbol{\theta}^*, \boldsymbol{\omega}^*\right) \leqslant \mathcal{O}_p\left(1/\sqrt{n}\right) \tag{10}$$

in our paper, where $n$ indicates the rough number of training data of CREs. Then, we provide the
excess risk bound of CREs.

**Lemma 2** (Bartlett et al. (2006)). *If the loss function $\ell(\cdot, \cdot)$ is non-negative classification-calibrated,
there exists a convex, non-decreasing, and invertible function $\varphi : [0, 1] \to [0, +\infty)$ such that for
any sequence $(u_i) \in [0, 1]$, $\varphi(u_i) \to 0$ if and only if $u_i \to 0$, and for any measurable function $f$
and $g$, and probability distribution over $\mathcal{X} \times \mathcal{Y}$,*

$$\varphi\left(R_{0-1}\left(\boldsymbol{\theta}, \boldsymbol{\omega}\right) - R_{0-1}^*\right) \leqslant R\left(\boldsymbol{\theta}, \boldsymbol{\omega}\right) - R^*. \tag{11}$$

Then, we provide an upper bound for the expected misclassification rate of $g_1$.

**Lemma 3.** *Assuming that the Vision Transformer used for fine-tuning is very flexible, for any $\delta > 0$,
the following inequality holds with probability $1 - \delta$, we have*

$$R_{0-1}\left(\widehat{\boldsymbol{\theta}}_1, \widehat{\boldsymbol{\omega}}_1\right) \leqslant R_{0-1}^* + \varphi^{-1}\left(\mathcal{O}_p\left(1/\sqrt{n}\right)\right) \tag{12}$$

*Proof.* With probability $1 - \delta$, we have

$$\begin{aligned}
R_{0-1}\left(\widehat{\boldsymbol{\theta}}_1, \widehat{\boldsymbol{\omega}}_1\right) - R_{0-1}^* &\leqslant \varphi^{-1}\left(R\left(\widehat{\boldsymbol{\theta}}_1, \widehat{\boldsymbol{\omega}}_1\right) - R^*\right) \\
&= \varphi^{-1}\left(R\left(\widehat{\boldsymbol{\theta}}_1, \widehat{\boldsymbol{\omega}}_1\right) - R\left(\boldsymbol{\theta}^*, \boldsymbol{\omega}^*\right) + R\left(\boldsymbol{\theta}^*, \boldsymbol{\omega}^*\right) - R^*\right) \\
&= \varphi^{-1}\left(R\left(\widehat{\boldsymbol{\theta}}_1, \widehat{\boldsymbol{\omega}}_1\right) - R\left(\boldsymbol{\theta}^*, \boldsymbol{\omega}^*\right)\right) \\
&\leqslant \varphi^{-1}\left(\mathcal{O}_p\left(1/\sqrt{n}\right)\right).
\end{aligned}$$

Here, the first inequality is due to Lemma 2, the second equality is because we have $R\left(\boldsymbol{\theta}^*, \boldsymbol{\omega}^*\right) = R^*$ when the model is very flexible. The proof is completed. □

Finally, we give the proof of Theorem 1.

*Proof of Theorem 1.* On one hand, we have

$$\widehat{R}_2\left(\boldsymbol{\theta}, \boldsymbol{\omega}\right)$$

$$=\frac{\pi_{\mathrm{Te}}}{\left|\widehat{D}_+\right|} \sum_{i \in \widehat{D}_+} \ell\left(g\left(\boldsymbol{f}\left(\boldsymbol{x}_i\right)\right), +1\right) + \frac{1 - \pi_{\mathrm{Te}}}{\left|\widehat{D}_-\right|} \sum_{j \in \widehat{D}_-} \ell\left(g\left(\boldsymbol{f}\left(\boldsymbol{x}_j\right)\right), -1\right)$$

$$=\sum_{i=1}^{\left|\widehat{D}_{\mathrm{W}}\right|} \left(\mathbb{I}\left[\sigma\left(g_1\left(\boldsymbol{f}\left(\boldsymbol{x}_i\right)\right)\right) > 0.5\right] \frac{\pi_{\mathrm{Te}}}{\left|\widehat{D}_+\right|} \ell\left(g\left(\boldsymbol{f}\left(\boldsymbol{x}_i\right)\right), +1\right)\right.$$

$$\left. + \mathbb{I}\left[\sigma\left(g_1\left(\boldsymbol{f}\left(\boldsymbol{x}_i\right)\right)\right) < 0.5\right] \frac{1 - \pi_{\mathrm{Te}}}{\left|\widehat{D}_-\right|} \ell\left(g\left(\boldsymbol{f}\left(\boldsymbol{x}_j\right)\right), -1\right)\right)$$

$$\leqslant \sum_{i=1}^{\left|\widehat{D}_{\mathrm{W}}\right|} \left(\left(\mathbb{I}\left[y_i = +1\right] + \mathbb{I}\left[y_i = -1, \sigma\left(g_1\left(\boldsymbol{f}\left(\boldsymbol{x}_i\right)\right)\right) > 0.5\right]\right) \frac{\pi_{\mathrm{Te}}}{\left|\widehat{D}_+\right|} \ell\left(g\left(\boldsymbol{f}\left(\boldsymbol{x}_i\right)\right), +1\right)\right.$$

$$\left. + \left(\mathbb{I}\left[y_i = -1\right] + \mathbb{I}\left[y_i = +1, \sigma\left(g_1\left(\boldsymbol{f}\left(\boldsymbol{x}_i\right)\right)\right) < 0.5\right]\right) \frac{1 - \pi_{\mathrm{Te}}}{\left|\widehat{D}_-\right|} \ell\left(g\left(\boldsymbol{f}\left(\boldsymbol{x}_j\right)\right), -1\right)\right)$$

$$=\frac{1}{\left|\widehat{D}_{\mathrm{W}}\right|} \sum_{i=1}^{\left|\widehat{D}_{\mathrm{W}}\right|} \left(\mathbb{I}\left[y_i = +1\right] \ell\left(g\left(\boldsymbol{f}\left(\boldsymbol{x}_i\right)\right), +1\right) + \mathbb{I}\left[y_i = -1\right] \ell\left(g\left(\boldsymbol{f}\left(\boldsymbol{x}_i\right)\right), -1\right)\right)$$

$$+ \frac{1}{\left|\widehat{D}_{\mathrm{W}}\right|} \sum_{i=1}^{\left|\widehat{D}_{\mathrm{W}}\right|} \left(\mathbb{I}\left[y_i = -1, \sigma\left(g_1\left(\boldsymbol{f}\left(\boldsymbol{x}_i\right)\right)\right) > 0.5\right] \ell\left(g\left(\boldsymbol{f}\left(\boldsymbol{x}_i\right)\right), +1\right)\right.$$

$$+ \mathbb{I}\left[y_i = +1, \sigma\left(g_1\left(\boldsymbol{f}\left(\boldsymbol{x}_i\right)\right)\right) < 0.5\right] \ell\left(g\left(\boldsymbol{f}\left(\boldsymbol{x}_i\right)\right), -1\right)$$

$$\leqslant \frac{1}{\left|\widehat{D}_{\mathrm{W}}\right|} \sum_{i=1}^{\left|\widehat{D}_{\mathrm{W}}\right|} \ell\left(g\left(\boldsymbol{f}\left(\boldsymbol{x}_i\right)\right), y_i\right)$$

$$+ \frac{C_\ell}{\left|\widehat{D}_{\mathrm{W}}\right|} \sum_{i=1}^{\left|\widehat{D}_{\mathrm{W}}\right|} \left(\mathbb{I}\left[y_i = -1, \sigma\left(g_1\left(\boldsymbol{f}\left(\boldsymbol{x}_i\right)\right)\right) > 0.5\right] + \mathbb{I}\left[y_i = +1, \sigma\left(g_1\left(\boldsymbol{f}\left(\boldsymbol{x}_i\right)\right)\right) < 0.5\right]\right).$$

Therefore, with probability $1 - \delta$, we have

$$\mathbb{E}_{\widehat{D}_{\mathrm{W}}}\left[\widehat{R}_2\left(\boldsymbol{\theta}, \boldsymbol{\omega}\right)\right] \leqslant R\left(\boldsymbol{\theta}, \boldsymbol{\omega}\right) + C_\ell\left(R_{0-1}\left(\widehat{\boldsymbol{\theta}}_1, \widehat{\boldsymbol{\omega}}_1\right)\right)$$

$$\leqslant R\left(\boldsymbol{\theta}, \boldsymbol{\omega}\right) + C_\ell\left(R_{0-1}^* + \varphi^{-1}\left(\mathcal{O}_p\left(1/\sqrt{n}\right)\right)\right).$$

On the other hand, we have

$$\widehat{R}_2\left(\boldsymbol{\theta}, \boldsymbol{\omega}\right)$$

$$= \frac{\pi_{\mathrm{Te}}}{\left|\widehat{D}_+\right|} \sum_{i \in \widehat{D}_+} \ell\left(g\left(\boldsymbol{f}\left(\boldsymbol{x}_i\right)\right), +1\right) + \frac{1 - \pi_{\mathrm{Te}}}{\left|\widehat{D}_-\right|} \sum_{j \in \widehat{D}_-} \ell\left(g\left(\boldsymbol{f}\left(\boldsymbol{x}_j\right)\right), -1\right)$$

$$= \sum_{i=1}^{\left|\widehat{D}_{\mathrm{W}}\right|} \left(\mathbb{I}\left[\sigma\left(g_1\left(\boldsymbol{f}\left(\boldsymbol{x}_i\right)\right)\right) > 0.5\right] \frac{\pi_{\mathrm{Te}}}{\left|\widehat{D}_+\right|} \ell\left(g\left(\boldsymbol{f}\left(\boldsymbol{x}_i\right)\right), +1\right)\right.$$

$$\left. + \mathbb{I}\left[\sigma\left(g_1\left(\boldsymbol{f}\left(\boldsymbol{x}_i\right)\right)\right) < 0.5\right] \frac{1 - \pi_{\mathrm{Te}}}{\left|\widehat{D}_-\right|} \ell\left(g\left(\boldsymbol{f}\left(\boldsymbol{x}_j\right)\right), -1\right)\right)$$

$$\geqslant \sum_{i=1}^{\left|\widehat{D}_{\mathrm{W}}\right|} \left(\mathbb{I}\left[y_i = +1, \sigma\left(g_1\left(\boldsymbol{f}\left(\boldsymbol{x}_i\right)\right)\right) > 0.5\right] \frac{\pi_{\mathrm{Te}}}{\left|\widehat{D}_+\right|} \ell\left(g\left(\boldsymbol{f}\left(\boldsymbol{x}_i\right)\right), +1\right)\right.$$

$$\left. + \mathbb{I}\left[y_i = -1, \sigma\left(g_1\left(\boldsymbol{f}\left(\boldsymbol{x}_i\right)\right)\right) < 0.5\right] \frac{1 - \pi_{\mathrm{Te}}}{\left|\widehat{D}_-\right|} \ell\left(g\left(\boldsymbol{f}\left(\boldsymbol{x}_j\right)\right), -1\right)\right)$$

$$= \frac{1}{\left|\widehat{D}_{\mathrm{W}}\right|} \sum_{i=1}^{\left|\widehat{D}_{\mathrm{W}}\right|} \left(\mathbb{I}\left[y_i = +1\right] \ell\left(g\left(\boldsymbol{f}\left(\boldsymbol{x}_i\right)\right), +1\right) + \mathbb{I}\left[y_i = -1\right] \ell\left(g\left(\boldsymbol{f}\left(\boldsymbol{x}_i\right)\right), -1\right)\right)$$

$$- \frac{1}{\left|\widehat{D}_{\mathrm{W}}\right|} \sum_{i=1}^{\left|\widehat{D}_{\mathrm{W}}\right|} \left(\mathbb{I}\left[y_i = -1, \sigma\left(g_1\left(\boldsymbol{f}\left(\boldsymbol{x}_i\right)\right)\right) > 0.5\right] \ell\left(g\left(\boldsymbol{f}\left(\boldsymbol{x}_i\right)\right), +1\right)\right.$$

$$\left. - \mathbb{I}\left[y_i = +1, \sigma\left(g_1\left(\boldsymbol{f}\left(\boldsymbol{x}_i\right)\right)\right) < 0.5\right] \ell\left(g\left(\boldsymbol{f}\left(\boldsymbol{x}_i\right)\right), -1\right)\right)$$

$$\geqslant \frac{1}{\left|\widehat{D}_{\mathrm{W}}\right|} \sum_{i=1}^{\left|\widehat{D}_{\mathrm{W}}\right|} \ell\left(g\left(\boldsymbol{f}\left(\boldsymbol{x}_i\right)\right), y_i\right)$$

$$- \frac{C_\ell}{\left|\widehat{D}_{\mathrm{W}}\right|} \sum_{i=1}^{\left|\widehat{D}_{\mathrm{W}}\right|} \left(\mathbb{I}\left[y_i = -1, \sigma\left(g_1\left(\boldsymbol{f}\left(\boldsymbol{x}_i\right)\right)\right) > 0.5\right] + \mathbb{I}\left[y_i = +1, \sigma\left(g_1\left(\boldsymbol{f}\left(\boldsymbol{x}_i\right)\right)\right) < 0.5\right]\right).$$

Therefore, the following inequality holds with probability $1 - \delta$:

$$\mathbb{E}_{\widehat{D}_{\mathrm{W}}}\left[\widehat{R}_2\left(\boldsymbol{\theta}, \boldsymbol{\omega}\right)\right] - R\left(\boldsymbol{\theta}, \boldsymbol{\omega}\right) \geqslant -C_\ell\left(R_{0-1}^* + \varphi^{-1}\left(\mathcal{O}_p\left(1/\sqrt{n}\right)\right)\right).$$

Therefore, we have the following inequality with probability $1 - 2\delta$:

$$\left|R_2\left(\boldsymbol{\theta}, \boldsymbol{\omega}\right) - R\left(\boldsymbol{\theta}, \boldsymbol{\omega}\right)\right| \leqslant C_\ell\left(R_{0-1}^* + \varphi^{-1}\left(\mathcal{O}_p\left(1/\sqrt{n}\right)\right)\right), \tag{13}$$

which concludes the proof. $\qquad\square$

### F.2 PROOF OF THEOREM 2

**Definition 1** (Rademacher Complexity). Let $\widehat{D}_+ = \{\widehat{\boldsymbol{x}}_1^+, \cdots \widehat{\boldsymbol{x}}_{n_+}^+\}$ denote $n_+$ i.i.d. random variables drawn from a probability distribution with the density $\widehat{p}_+(\boldsymbol{x})$, and let $\widehat{D}_- = \{\widehat{\boldsymbol{x}}_1^-, \cdots \widehat{\boldsymbol{x}}_{n_-}^-\}$ denote $n_-$ i.i.d. random variables drawn from a probability distribution with the density $\widehat{p}_-(\boldsymbol{x})$. Let $\Phi = \{f \circ g : \mathcal{X} \mapsto \mathbb{R}\}$ denote a class of measurable functions that includes the encoder and the classifier, and $\boldsymbol{\sigma} = (\sigma_1, \sigma_2, \cdots, \sigma_n)$ denote Rademacher variables taking values from $\{+1, -1\}$

uniformly. Then, the (expected) Rademacher complexity of $\Phi$ w.r.t. $\widehat{D}_+$ and $\widehat{D}_-$ is defined as

$$\mathfrak{R}_{n_+}(\Phi) = \mathbb{E}_{\widehat{D}_+} \mathbb{E}_{\boldsymbol{\sigma}} \left[ \sup_{\phi \in \Phi} \frac{1}{n_+} \sum_{i=1}^{n_+} \sigma_i \phi\left(\widehat{\boldsymbol{x}}_i^+\right) \right],$$

$$\mathfrak{R}'_{n_-}(\Phi) = \mathbb{E}_{\widehat{D}_-} \mathbb{E}_{\boldsymbol{\sigma}} \left[ \sup_{\phi \in \Phi} \frac{1}{n_-} \sum_{i=1}^{n_-} \sigma_i \phi\left(\widehat{\boldsymbol{x}}_i^-\right) \right].$$

**Lemma 4.** *For any $\delta > 0$, we have the following inequality with probability at least $1 - \delta$:*

$$\sup_{\boldsymbol{\theta} \in \Theta, \boldsymbol{\omega} \in \Omega} \left| \widehat{R}_2(\boldsymbol{\theta}, \boldsymbol{\omega}) - R_2(\boldsymbol{\theta}, \boldsymbol{\omega}) \right| \leqslant 2\pi_{\text{Te}} L_\ell \mathfrak{R}_{n_+}(\Phi) + 2\left(1 - \pi_{\text{Te}}\right) L_\ell \mathfrak{R}'_{n_-}(\Phi)$$

$$+ \left( \frac{\pi_{\text{Te}} C_\ell}{\sqrt{n_+}} + \frac{\left(1 - \pi_{\text{Te}}\right) C_\ell}{\sqrt{n_-}} \right) \sqrt{\frac{\ln 2/\delta}{2}}.$$

*Proof.* To begin with, we investigate the upper bound for the one-side uniform deviation $\sup_{\boldsymbol{\theta} \in \Theta, \boldsymbol{\omega} \in \Omega} \left( \widehat{R}_2(\boldsymbol{\theta}, \boldsymbol{\omega}) - R_2(\boldsymbol{\theta}, \boldsymbol{\omega}) \right)$. When an example in $\widehat{D}_+$ is replaced by another example, the value of $\sup_{\boldsymbol{\theta} \in \Theta, \boldsymbol{\omega} \in \Omega} \left( \widehat{R}_2(\boldsymbol{\theta}, \boldsymbol{\omega}) - R_2(\boldsymbol{\theta}, \boldsymbol{\omega}) \right)$ changes at most $\pi_{\text{Te}} C_\ell / n_+$. When a example in $\widehat{D}_-$ is replaced by another example, the value of $\sup_{\boldsymbol{\theta} \in \Theta, \boldsymbol{\omega} \in \Omega} \left( \widehat{R}_2(\boldsymbol{\theta}, \boldsymbol{\omega}) - R_2(\boldsymbol{\theta}, \boldsymbol{\omega}) \right)$ changes at most $\left(1 - \pi_{\text{Te}}\right) C_\ell / n_-$. Therefore, according to McDiarmid's inequality, we have that

$$p \left( \sup_{\boldsymbol{\theta} \in \Theta, \boldsymbol{\omega} \in \Omega} \left( \widehat{R}_2(\boldsymbol{\theta}, \boldsymbol{\omega}) - R_2(\boldsymbol{\theta}, \boldsymbol{\omega}) \right) - \mathbb{E} \left[ \sup_{\boldsymbol{\theta} \in \Theta, \boldsymbol{\omega} \in \Omega} \left( \widehat{R}_2(\boldsymbol{\theta}, \boldsymbol{\omega}) - R_2(\boldsymbol{\theta}, \boldsymbol{\omega}) \right) \right] \geqslant \epsilon \right)$$

$$\leqslant \exp \left( \frac{-2\epsilon^2}{\frac{\pi_{\text{Te}}^2 C_\ell^2}{n_+} + \frac{(1-\pi_{\text{Te}})^2 C_\ell^2}{n_-}} \right).$$

In an equivalent way, we have the following inequality with probability at least $1 - \delta/2$:

$$\sup_{\boldsymbol{\theta} \in \Theta, \boldsymbol{\omega} \in \Omega} \left( \widehat{R}_2(\boldsymbol{\theta}, \boldsymbol{\omega}) - R_2(\boldsymbol{\theta}, \boldsymbol{\omega}) \right)$$

$$\leqslant \mathbb{E} \left[ \sup_{\boldsymbol{\theta} \in \Theta, \boldsymbol{\omega} \in \Omega} \left( \widehat{R}_2(\boldsymbol{\theta}, \boldsymbol{\omega}) - R_2(\boldsymbol{\theta}, \boldsymbol{\omega}) \right) \right] + \sqrt{\left( \frac{\pi_{\text{Te}}^2 C_\ell^2}{n_+} + \frac{\left(1 - \pi_{\text{Te}}\right)^2 C_\ell^2}{n_-} \right)} \sqrt{\frac{\ln 2/\delta}{2}}$$

$$\leqslant \mathbb{E} \left[ \sup_{\boldsymbol{\theta} \in \Theta, \boldsymbol{\omega} \in \Omega} \left( \widehat{R}_2(\boldsymbol{\theta}, \boldsymbol{\omega}) - R_2(\boldsymbol{\theta}, \boldsymbol{\omega}) \right) \right] + \left( \frac{\pi_{\text{Te}} C_\ell}{\sqrt{n_+}} + \frac{\left(1 - \pi_{\text{Te}}\right) C_\ell}{\sqrt{n_-}} \right) \sqrt{\frac{\ln 2/\delta}{2}}, \qquad (14)$$

where the last inequality is due to $\sqrt{a^2 + b^2} \leqslant |a| + |b|$. Besides, by symmetrization (Vapnik, 1998), it is a routine work to have that

$$\mathbb{E} \left[ \sup_{\boldsymbol{\theta} \in \Theta, \boldsymbol{\omega} \in \Omega} \left( \widehat{R}_2(\boldsymbol{\theta}, \boldsymbol{\omega}) - R_2(\boldsymbol{\theta}, \boldsymbol{\omega}) \right) \right] \leqslant 2\pi_{\text{Te}} \mathfrak{R}_{n_+}(\ell \circ \Phi) + 2\left(1 - \pi_{\text{Te}}\right) \mathfrak{R}'_{n_-}(\ell \circ \Phi). \qquad (15)$$

According to Talagrand's contraction lemma (Shalev-Shwartz & Ben-David, 2014), we have

$$\mathfrak{R}_{n_+}(\ell \circ \Phi) \leqslant L_\ell \mathfrak{R}_{n_+}(\Phi), \mathfrak{R}'_{n_-}(\ell \circ \Phi) \leqslant L_\ell \mathfrak{R}'_{n_-}(\Phi). \qquad (16)$$

Combining Inequalities 14, 15, and 16, we have the following inequality with probability at least $1 - \delta/2$:

$$\sup_{\boldsymbol{\theta} \in \Theta, \boldsymbol{\omega} \in \Omega} \left( \widehat{R}_2(\boldsymbol{\theta}, \boldsymbol{\omega}) - R_2(\boldsymbol{\theta}, \boldsymbol{\omega}) \right) \leqslant 2\pi_{\text{Te}} L_\ell \mathfrak{R}_{n_+}(\Phi) + 2\left(1 - \pi_{\text{Te}}\right) L_\ell \mathfrak{R}'_{n_-}(\Phi)$$

$$+ \left( \frac{\pi_{\text{Te}} C_\ell}{\sqrt{n_+}} + \frac{\left(1 - \pi_{\text{Te}}\right) C_\ell}{\sqrt{n_-}} \right) \sqrt{\frac{\ln 2/\delta}{2}}. \qquad (17)$$

In a similar way, we have the following inequality with probability at least $1 - \delta/2$:

$$\sup_{\boldsymbol{\theta} \in \Theta, \boldsymbol{\omega} \in \Omega} \left( R_2 \left( \boldsymbol{\theta}, \boldsymbol{\omega} \right) - \widehat{R}_2 \left( \boldsymbol{\theta}, \boldsymbol{\omega} \right) \right) \leqslant 2\pi_{\mathrm{Te}} L_\ell \mathfrak{R}_{n_+} \left( \Phi \right) + 2 \left( 1 - \pi_{\mathrm{Te}} \right) L_\ell \mathfrak{R}'_{n_-} \left( \Phi \right)$$

$$+ \left( \frac{\pi_{\mathrm{Te}} C_\ell}{\sqrt{n_+}} + \frac{\left( 1 - \pi_{\mathrm{Te}} \right) C_\ell}{\sqrt{n_-}} \right) \sqrt{\frac{\ln 2/\delta}{2}}. \tag{18}$$

Combining Inequalities 17 and 18, the proof is completed. $\qquad \square$

*Proof of Theorem 2.* We have the following inequality:

$$R_2 \left( \widehat{\boldsymbol{\theta}}_2, \widehat{\boldsymbol{\omega}}_2 \right) - R_2 \left( \boldsymbol{\theta}_2^*, \boldsymbol{\omega}_2^* \right)$$

$$= \left( R_2 \left( \widehat{\boldsymbol{\theta}}_2, \widehat{\boldsymbol{\omega}}_2 \right) - \widehat{R}_2 \left( \widehat{\boldsymbol{\theta}}_2, \widehat{\boldsymbol{\omega}}_2 \right) \right) + \left( \widehat{R}_2 \left( \widehat{\boldsymbol{\theta}}_2, \widehat{\boldsymbol{\omega}}_2 \right) - \widehat{R}_2 \left( \boldsymbol{\theta}_2^*, \boldsymbol{\omega}_2^* \right) \right) + \left( \widehat{R}_2 \left( \boldsymbol{\theta}_2^*, \boldsymbol{\omega}_2^* \right) - R_2 \left( \boldsymbol{\theta}_2^*, \boldsymbol{\omega}_2^* \right) \right)$$

$$\leqslant 0 + 2 \sup_{\boldsymbol{\theta} \in \Theta, \boldsymbol{\omega} \in \Omega} \left| \widehat{R}_2 \left( \boldsymbol{\theta}, \boldsymbol{\omega} \right) - R_2 \left( \boldsymbol{\theta}, \boldsymbol{\omega} \right) \right|$$

$$\leqslant 4\pi_{\mathrm{Te}} L_\ell \mathfrak{R}_{n_+} \left( \Phi \right) + 4 \left( 1 - \pi_{\mathrm{Te}} \right) L_\ell \mathfrak{R}'_{n_-} \left( \Phi \right) + \left( \frac{2\pi_{\mathrm{Te}} C_\ell}{\sqrt{n_+}} + \frac{\left( 2 - 2\pi_{\mathrm{Te}} \right) C_\ell}{\sqrt{n_-}} \right) \sqrt{\frac{\ln 2/\delta}{2}}.$$

Since we have $\mathfrak{R}_{n_+} \left( \Phi \right) \leqslant \mathcal{O}_p \left( 1/\sqrt{n_+} \right)$ and $\mathfrak{R}'_{n_-} \left( \Phi \right) \leqslant \mathcal{O}_p \left( 1/\sqrt{n_-} \right)$, the proof is completed. $\quad \square$

## G  A BRIEF INTRODUCTION TO SEVERAL WSL PROBLEMS

In this section, we provide a brief introduction to several representative WSL problems and their corresponding URE and CREs.

### G.1  PU LEARNING

In PU learning, we are given a positive training dataset $\mathcal{D}_{\mathrm{P}} = \{(\mathbf{x}_i^{\mathrm{P}}, +1)\}_{i=1}^{n_{\mathrm{P}}}$ and an unlabeled dataset $\mathcal{D}_{\mathrm{U}} = \{\mathbf{x}_i^{\mathrm{U}}\}_{i=1}^{n_{\mathrm{U}}}$ sampled from densities $p_+(\boldsymbol{x})$ and $p(\boldsymbol{x}) = \pi_{\mathrm{Te}} p_+(\boldsymbol{x}) + (1 - \pi_{\mathrm{Te}}) p_-(\boldsymbol{x})$ respectively. Here, $p(\boldsymbol{x})$ denotes the density of test data. du Plessis et al. (2015) proposed an URE for PU learning, i.e.,

$$\widehat{R}_{\mathrm{PU}}(\boldsymbol{\theta}, \boldsymbol{\omega}) = \frac{\pi_{\mathrm{Te}}}{n_{\mathrm{P}}} \sum_{i=1}^{n_{\mathrm{P}}} \left( \ell \left( g \left( \boldsymbol{f} \left( \mathbf{x}_i^{\mathrm{P}} \right) \right), +1 \right) - \ell \left( g \left( \boldsymbol{f} \left( \mathbf{x}_i^{\mathrm{P}} \right) \right), -1 \right) \right)$$

$$+ \frac{1}{n_{\mathrm{U}}} \sum_{i=1}^{n_{\mathrm{U}}} \ell \left( g \left( \boldsymbol{f} \left( \mathbf{x}_i^{\mathrm{U}} \right) \right), -1 \right). \tag{19}$$

Then, Kiryo et al. (2017) proposed a CRE to improve the classification performance:

$$\widehat{R}_{\mathrm{CPU}}(\boldsymbol{\theta}, \boldsymbol{\omega}) = \frac{\pi_{\mathrm{Te}}}{n_{\mathrm{P}}} \sum_{i=1}^{n_{\mathrm{P}}} \ell \left( g \left( \boldsymbol{f} \left( \mathbf{x}_i^{\mathrm{P}} \right) \right), +1 \right)$$

$$+ h \left( \frac{1}{n_{\mathrm{U}}} \sum_{i=1}^{n_{\mathrm{U}}} \ell \left( g \left( \boldsymbol{f} \left( \mathbf{x}_i^{\mathrm{U}} \right) \right), -1 \right) - \frac{\pi_{\mathrm{Te}}}{n_{\mathrm{P}}} \sum_{i=1}^{n_{\mathrm{P}}} \ell \left( g \left( \boldsymbol{f} \left( \mathbf{x}_i^{\mathrm{P}} \right) \right), -1 \right) \right). \tag{20}$$

### G.2  PCOMP LEARNING

Feng et al. (2021) investigated a problem called Pcomp learning. In this problem, we are given unlabeled training data pairs $\mathcal{D}_{\mathrm{Pcomp}} = \{(\mathbf{x}_i^{\mathrm{PC}}, \mathbf{x}_i^{\mathrm{PC}'})\}_{i=1}^{n_{\mathrm{PC}}}$ from (+1,+1), (+1,-1), and (-1,-1) are

given. It was shown that the problem can be transformed to UU learning (Lu et al., 2019) and an URE was proposed accordingly:

$$\widehat{R}_{\mathrm{Pcomp}}(\boldsymbol{\theta}, \boldsymbol{\omega}) = \frac{1}{n_{\mathrm{PC}}} \sum_{i=1}^{n_{\mathrm{PC}}} \Big( \ell(g(\boldsymbol{f}(\mathbf{x}_i^{\mathrm{PC}})), +1) + \ell(g(\boldsymbol{f}(\mathbf{x}_i^{\mathrm{PC}'})), -1)$$

$$- \pi_+ \ell(g(\boldsymbol{f}(\mathbf{x}_i^{\mathrm{PC}})), -1) - \pi_- \ell(g(\boldsymbol{f}(\mathbf{x}_i^{\mathrm{PC}'})), +1) \Big). \tag{21}$$

Then, a CRE was proposed as

$$\widehat{R}_{\mathrm{Pcomp}}(\boldsymbol{\theta}, \boldsymbol{\omega}) = h \left( \frac{1}{n_{\mathrm{PC}}} \sum_{i=1}^{n_{\mathrm{PC}}} \Big( \ell(g(\boldsymbol{f}(\mathbf{x}_i^{\mathrm{PC}})), +1) - \pi_- \ell(g(\boldsymbol{f}(\mathbf{x}_i^{\mathrm{PC}'})), +1) \Big) \right)$$

$$+ h \left( \frac{1}{n_{\mathrm{PC}}} \sum_{i=1}^{n_{\mathrm{PC}}} \Big( \ell(g(\boldsymbol{f}(\mathbf{x}_i^{\mathrm{PC}'})), -1) - \pi_+ \ell(g(\boldsymbol{f}(\mathbf{x}_i^{\mathrm{PC}})), -1) \Big) \right). \tag{22}$$

