# OpenReview forum: "Delving into Weakly Supervised Learning with Pre-Trained Models"
_ICLR.cc/2025/Conference — Submitted to ICLR 2025_

### Official Review · Reviewer_WU3A · 2024-10-28

**Soundness:** 3
**Presentation:** 2
**Contribution:** 3
**Rating:** 5
**Confidence:** 3

**Summary:**

This paper proposes a WSFT approach to effectively fine-tune pre-trained models for various WSL problems. WSFT encapsulates reliable supervision distillation and efficient model fine-tuning seamlessly supported by theoretical guarantees. Extensive experiments on various WSL problems and benchmark datasets validate the effectiveness of WSFT.

**Strengths:**

This paper proposes a novel weakly supervised fine-tuning approach using dual classification heads that are trained synergistically
by alternately distilling reliable supervision and performing efficient model fine-tuning. Theoretically, the authors prove the consistency and convergence rate of the proposed risk estimator.

**Weaknesses:**

1. The zero-shot baselines exhibit better performance in few pairs setting in table 2, which brings the concerns of effectiveness of WSFT in few shot scenarios. It would be better to provide additional analysis or experiments to examine WSFT's performance in other few pairs settings.
2. More recent methods are expected in comparison. Most of the compared apporaches seems to be out-of-data (2021 or before in table 2 and table 3).

**Questions:**

1. In table 1, the results of WSFT on Oxford-IIIT Pet-b is worse than CVIR and DistPU. Further analysis is expected to explain this phenomenon.
2. In table 2 and table 3, experiment results with more recent approaches should be presented.

---

> ### Author Response · Authors · 2024-11-23
> **Response to Reviewer WU3A**
>
> Thank you for taking the time and effort to review our paper. Please find below our responses to your comments and questions.
>
> **Q1: The zero-shot baselines exhibit better performance.**
>
> **A1:** We would like to respectfully point out that the zero-shot baseline uses both visual and textual information for classification, while WSFT and other WSL methods only train the vision encoder of CLIP using images without textual information. Therefore, in some rare cases where the training number is very small, it is reasonable that WSFT and other baselines may have relatively lower performance than zero-shot CLIP on some datasets.
>
> Note that other VLM fine-tuning works, such as CoOp [1], also have lower performance than the zero-shot baseline on some datasets when the number of samples is small. It is also worth noting that in these cases WSFT still outperforms all existing WSL methods by a large margin.
>
> **Q2: More recent methods are expected in comparison.**
>
> **A2:** We would like to respectfully point out that we have included almost all of the recent baselines in the paper. There are no baselines for UU and Pcomp learning from 2021 to 2023 that we are aware of. Following GLWS, we have included all the baselines used in the GLWS paper for UU and Pcomp learning in our study. In addition, we have included the results of the most recent and state-of-the-art method GLWS published in ICML 2024 in our experimental comparison. Therefore, we consider the baselines to be comprehensive and up-to-date.
>
> **Q3: Analysis about the results of WSFT on Oxford-IIIT Pet-b is worse than CVIR and DistPU.**
>
> **A3:** We would like to respectfully point out that WSFT significantly outperforms all existing methods, except for *only one case* when the number of positive/unknown samples is 180/2220 in the Oxford-IIIT Pet-b dataset for PU learning.
>
> The effectiveness of these two methods may be due to their compatibility with relatively simple datasets, as evidenced by the high performance of zero-shot CLIP, and when there is a significant amount of positive data. Under these conditions, the direct application of pseudo-labeling for training (CVIR) or the use of data distribution information for training (DistPU) tends to yield good results. However, their performance degrades as the amount of positive data decreases and the dataset or task becomes more challenging.
>
> In particular, according to the "no free lunch" theorem in machine learning, we cannot expect that there is a machine learning model that can win in all cases on all datasets. The experimental results are sufficient to validate the superiority of our approach over state-of-the-art methods.
>
> **Q4: In table 2 and table 3, experiment results with more recent approaches should be presented.**
>
> **A4:** Please see A2.
>
> Reference:
>
> [1] Learning to Prompt for Vision-Language Models, IJCV 2022.

---

> ### Author Response · Authors · 2024-11-28
>
> Dear Reviewer WU3A，
>
> We would like to sincerely thank you for your efforts and comments in reviewing the submission. As we approach the end of the rebuttal period, we would like to ask you to kindly confirm that you have reviewed the rebuttal and let us know if there are any remaining concerns regarding our work. As mentioned in our previous response, we have clarified your questions about **zero-shot baselines**, **baseline comparison**, and **results analysis**. We hope that these responses will effectively address your questions. If you have any further concerns or questions, please do not hesitate to contact us. Thank you again for your help and time in reviewing our submission!
>
> Authors

---

### Official Review · Reviewer_p8yg · 2024-11-02

**Soundness:** 4
**Presentation:** 4
**Contribution:** 3
**Rating:** 6
**Confidence:** 4

**Summary:**

The authors proposed a WSFT method for effectively fine-tuning pre-trained models for various WSL problems. Specifically, WSFT uses dual classification heads to alternately combine the reliability of supervision distillation and the efficiency of model fine-tuning, and theoretically proves the correctness of its methods. Finally, the authors conducted extensive experiments on various WSL problems and benchmark datasets, validating the effectiveness of the proposed method compared to SOTA WSL methods.

**Strengths:**

The authors' method has a solid theoretical proof with a detailed proof process. The main theme of the article is clear. It first finds that the original weakly supervised training is not as good as the performance of the Clip model, then further analyzes the problem in depth, introduces new theories to solve it, and ultimately proves the correctness and effectiveness of the method through both theory and experiments.

**Weaknesses:**

An overview figure could be added to make the entire process clearer.

**Questions:**

(1) For binary classification tasks, is the conclusion related to the division of the dataset categories? How much impact does directly using GPT for division of the datasets have on the results, and has any verification been done for other combinations? How to consider the impact of the division of positive and negative samples on the results?

(2) Regarding question 1, can a simple verification of the effect on multi-class classification be conducted?

---

> ### Author Response · Authors · 2024-11-23
> **Response to Reviewer p8yg**
>
> Thanks for the reviewer's time and effort in reviewing our paper. We are encouraged that you agree with the contributions of our paper. Below are our responses to the reviewer's comments and questions.
>
> **Q1: An overview figure could be added to make the entire process clearer.**
>
> **A1:** Thank you for your suggestion. We have added an overview figure (Figure 5) in Appendix A due to the page limit.
>
> **Q2: Impact of the division of positive and negative samples on the results and results of other combinations.**
>
> **A2:** The separation of positive and negative groups can affect the difficulty of the task. For example, if the semantic distinction between the two groups is very clear, the model may learn more easily, resulting in better performance. Conversely, if the distinction is less clear, the model's performance may suffer. We verify this by using a different partitioning strategy on the CIFAR-100 dataset. In the paper, the partitioning is done into two groups: "living organisms and plants" versus "man-made objects, scenes, and people". We then asked GPT-4o to partition the classes in CIFAR-100 using a different method: "dynamic entities (mobile or potential for movement)" versus "static or stationary entities". The results of this partitioning in PU learning are shown in the table below.
>
> | uPU          | nnPU         | VarPU        | CVIR           |
> |--------------|--------------|--------------|----------------|
>  | 56.00±0.00   | 80.73±4.51   | 44.00±0.00   | 74.07±2.95     |
>
> |DistPU        | Count Loss    | GLWS          | Zero-Shot     | **WSFT**       |
> |---------------|---------------|---------------|---------------|----------------|
> |70.97±13.00   | 77.83±3.91    | 78.17±1.01    | 72.29±0.00    | **87.64±1.91** |
>
> Although this partition is much harder, WSFT still outperforms all existing methods by a wide margin, showing that WSFT is robust to different partitioning strategies. We have included these new results in Appendix D.
>
> **Q3: A simple verification of the effect on multi-class classification.**
>
> **A3:** Thanks for your suggestion. We have been conducting experiments on a multi-class weakly supervised learning task, and will present our results as soon as we finish them.

---

> ### Author Response · Authors · 2024-11-28
>
> Dear Reviewer p8yg，
>
> We would like to sincerely thank you for your efforts and comments in reviewing the submission. As we approach the end of the rebuttal period, we would like to ask you to kindly confirm that you have reviewed the rebuttal and let us know if there are any remaining concerns regarding our work. As mentioned in our previous response, in the revised paper, we have **added an overall pipeline figure** and **included the results of CIFAR-100 with a different partition**. **We will present our results on multi-class classification as soon as we finish them**. We hope that these responses will effectively address your questions. If you have any further concerns or questions, please do not hesitate to contact us.
>
> Thank you again for your help and time in reviewing our submission!
>
> Authors

---

### Official Review · Reviewer_tjXA · 2024-11-04

**Soundness:** 3
**Presentation:** 2
**Contribution:** 2
**Rating:** 3
**Confidence:** 3

**Summary:**

The paper investigates the potential of leveraging pre-trained models for weakly supervised learning (WSL).  The authors first demonstrate that a zero-shot baseline using a CLIP model with GPT-4o enriched class descriptions surpasses existing state-of-the-art WSL methods trained from scratch. Recognizing the potential for further improvement through fine-tuning, they identify limitations with existing unbiased and corrected risk estimators, noting issues with overfitting exacerbation and feature degeneration.

To address these issues, they propose a novel "Weakly Supervised Fine-Tuning (WSFT)" approach. WSFT utilizes dual classification heads: one trained on a corrected risk estimator to distill reliable supervision, and another trained via empirical risk minimization on the subset of data with high-confidence predictions from the first head. This iterative process allows for efficient model fine-tuning. The authors provide theoretical guarantees for the consistency and convergence rate of their proposed risk estimator. Finally, they present extensive empirical results on various WSL problems (PU learning, Pcomp learning, and UU learning) across benchmark datasets (CIFAR-100, EuroSAT, Oxford-IIIT Pet, Caltech-101), showing that WSFT consistently outperforms existing SOTA WSL approaches. In summary, the paper’s main contributions are: demonstrating the power of pre-trained models for WSL, proposing the novel WSFT algorithm with theoretical grounding, and empirically validating the effectiveness of WSFT across a range of WSL problems and datasets.

**Strengths:**

+ This paper tackles the largely unexplored intersection of pre-trained models and weakly supervised learning, showing the untapped potential of existing pre-trained models for WSL.

+ The paper is written clearly and concisely, with a logical flow that guides the reader through the motivation, methodology, theoretical analysis, and experimental results.

**Weaknesses:**

- While the paper presents extensive experiments across various benchmark datasets, the choice of datasets could be expanded to include more diverse and challenging scenarios, e.g., ImageNet and other variants.

- This paper does not explain how to generalize to multiple-class setting, especially what changes need to be made in the loss objective.

**Questions:**

- How robust is WSFT to varying levels of label noise and class imbalance?

- Given the connection to self-training, have you considered including a self-training baseline using the same pre-trained model and data augmentation strategies for a direct comparison?  This would help isolate the specific benefits of the dual head approach and supervision distillation in WSFT.

---

> ### Author Response · Authors · 2024-11-23
> **Response to Reviewer tjXA**
>
> First of all, we would like to express our sincere gratitude for the reviewer's efforts and time in reviewing our submission. Below are the responses to the reviewer's questions and comments.
>
> **Q1: The choice of datasets could be expanded to include more diverse and challenging scenarios, e.g., ImageNet and other variants.**
>
> **A1:** Thank you for your suggestion. We have performed further experiments on the ImageNet-100 dataset using PU learning. The results are shown in the following table. The proposed WSFT also outperforms all existing methods.
>
> | uPU        | nnPU       | VarPU      | CVIR       |
> |------------|------------|------------|------------|
> | 40.00±0.00 | 85.43±1.95 | 60.00±0.00 | 92.60±1.87 |
>
> | DistPU     | Count Loss | GLWS       | Zero-Shot | **WSFT**     |
> |------------|------------|------------|------------|------------|
> | 88.27±1.45 | 79.93±1.95 | 87.97±2.46 | 75.26±0.00 | **92.90±2.07** |
>
> We have included these results in Appendix D.
>
> **Q2: How to generalize to multiple-class setting?**
>
> **A2:** The major different is in Eq. (5) and line 251-252. First, We have
> \begin{equation}
> \widehat{D}\_{k}=\\{\boldsymbol{x}\_{i}|\boldsymbol{x}\_{i}\in\widehat{D}\_{\mathrm{W}},\max\_{k}(g\_{1k}(\boldsymbol{f}(\boldsymbol{x}\_{i})))>\tau\\}
> \end{equation}
> where $g\_{1k}$ is the modeling output of the first classification head for the $k$-th class and $\tau$ is a large threshold.
> Then, we Eq. (5) can be expressed as
> \begin{equation}
> \widehat{R}\_{2}(\boldsymbol{\theta},\boldsymbol{\omega}\_{2})=\sum\_{k}\frac{\pi\_{k}}{|\widehat{D}\_{k}|}\sum\_{\boldsymbol{x}\_i\in\widehat{D}\_{k}}\ell(g\_{2}(\boldsymbol{f}(\boldsymbol{x}\_{i})),k)
> \end{equation}
> where $\pi\_{k}$ is the class prior for the $k$-th class.
>
> **Q3: How robust is WSFT to varying levels of label noise and class imbalance?**
>
> **A3:** We would like to kindly point out that WSFT is robust to varying levels of label noise and class
> imbalance.
>
> - For label noise, it is important to note that in PU learning, a decrease in the number of labeled data often leads to a significant decline in the performance of other existing methods. In contrast, the performance of WSFT remains stable, as shown in Table 1. A smaller amount of labeled positive data means the model is trained with less supervised information, which can result in more noise in the unlabeled data. Therefore, WSFT demonstrates robustness to varying levels of label noise.
> - Regarding varying class imbalance, please note that we conducted experiments with varying class priors in both Pcom and UU learning. Moreover, in PU learning, the class prior varies significantly across datasets; for example, the class prior is 0.51 for Oxford-IIIT Pet and 0.7 for EuroSAT (Table 7). WSFT performs well across all these settings and datasets, outperforming existing methods. This demonstrates that WSFT is robust to different levels of class imbalance.
>
> **Q4: A self-training baseline using the same pre-trained model and data augmentation strategies.**
>
> **A4:** Thank you for your suggestion. We investigated such a baseline in our ablation study (Section 4.5). We conducted experiments using only a single head for self-training ("single head" in Table 5) to verify the effectiveness of dual heads and distillation. The results show that single head with self-training is inferior to WSFT, confirming the advantages of using dual heads and the proposed supervision distillation technique.

---

> ### Author Response · Authors · 2024-11-28
>
> Dear Reviewer tjXA，
>
> We would like to sincerely thank you for your efforts and comments in reviewing the submission. As we approach the end of the rebuttal period, we would like to ask you to kindly confirm that you have reviewed the rebuttal and let us know if there are any remaining concerns regarding our work. As mentioned in our previous response, we **have included results on ImageNet-100 in the revised paper** and clarified your questions about **how to generalize to the multiple-class setting**, **how robust is WSFT to varying levels of label noise and class imbalance**, and **self-training baselines**. We hope that these responses will effectively address your questions. If you have any further concerns or questions, please do not hesitate to contact us.
>
> Thank you again for your help and time in reviewing our submission!
>
> Authors

---

### Official Review · Reviewer_JYAz · 2024-11-05

**Soundness:** 4
**Presentation:** 1
**Contribution:** 2
**Rating:** 5
**Confidence:** 3

**Summary:**

This paper explores the zero-shot ability of CLIP on weakly-supervised learning. They propose a two-step algorithm for WSL by generating a pseudo label first. The experiments show improvement in various settings of WSL.

**Strengths:**

1. The author provides mathematical proof for their algorithm.
2. The experiments are detailed and show significant improvement to SOTA.

**Weaknesses:**

1. The writing is not clear.
2. Putting Sec. 3.1 in the approach section is questionable to me. See Q1.
3. Possible missing related work. See Q2.

**Questions:**

1. What is the difference between the zero-shot CLIP baseline introduced in L159 and the zero-shot performance on CIFAR-100 on the official CLIP GitHub repo (https://github.com/openai/CLIP)?

2. What is the author's comment on the comparison to [1]? To me, semantic segmentation, which can be viewed as a per-pixel image classification problem, is more complicated than image classification, which is the focus of this paper. It is mentioned that this paper can be extended to the "multi-class classification problem" in L276.

3. What is the formulation of PU learning? How is it compared to UU learning? This should be introduced before using PU learning as a major topic in L201. PU learning seems to be a special case for UU learning as mentioned in L223.

4. I am confused about Sec. 3.2, which is about the motivation of this paper. Does the depicted problem only occur in PU learning? In UU learning? Or both? The author explains the problem by giving PU learning as an example but later talks about UU learning in Sec. 3.3.

5. What is CRE $\hat R_1$ in L239? Is it the same as Eq 3, which is $\hat R_{CUU}$?

6. In L244, what does it mean "could contain all the unlabeled data whose true labels are not accessible to the learning algorithm"? Is there unlabeled data whose true labels are accessible? How can one access the labels of unlabeled data?

7. I think the writing of Sec. 3.3 can be improved. Especially, the definition of supervision distillation and model fine-tuning should be explained or summarized first in L237 at the start of the paragraph. Is the first step, i.e. supervision distillation, from L239 to L244, or L239 to L257? What is model-finetuning then?

8. Are the "class priors" in Sec. 4.3 and 4.4 the same as $\pi_{T_e}$?

9. The formulation of Pcomp should be explained somewhere, especially the meaning of "unlabeled pairwise observations were provided" in L402.

10. The legend in Fig. 2(d) is not visible.

11. The notation $C$ is first used in Eq. 2. The author also uses $C_g$ and $C_l$ for various constants in the theoretical analysis. I recommend distinguishing them. Additionally, $D$ is used in Eq. 2 and also as some datasets.

[1] CLIP is Also an Efficient Segmenter: A Text-Driven Approach for Weakly Supervised Semantic Segmentation. [CVPR 2023]

---

> ### Author Response · Authors · 2024-11-23
> **Response to Reviewer  JYAz (1/2)**
>
> Thank you for the reviewer's time and efforts in reviewing our paper. Below are the responses to the reviewer's questions or comments.
>
> **Q1: What is the difference between the zero-shot CLIP baseline introduced in L159 and the zero-shot performance on CIFAR-100 on the official CLIP GitHub repo?**
>
> **A1:** Since the problem setting is different, the experimental results are naturally different. In this paper, we mainly deal with binary classification, while the original CLIP is for multi-class classification. Hence, the experimental results are different. We refer the reader to Section 3.1 for a detailed introduction to the zero-shot baseline used in this paper, and to Appendix A for the algorithmic details.
>
> **Q2: What is the author's comment on the comparison to Lin et al.?**
>
> **A2:** The problem studied and even the research area is completely different. We are working on **weakly supervised machine learning**, a research topic in the field of machine learning, please refer to [1], [2], and [10] to know our research background. We understand the difficulty and importance of weakly supervised semantic segmentation, but it is for computer vision and has no relation to our work. In fact, PU learning is also important and has a wide range of applications in machine learning. **We respectfully disagree that a machine learning paper should be considered less useful because the scale of the problem is smaller than a similarly named problem in computer vision, a completely different community.** We have cited [1] and discussed the difference in the revised version of our paper.
>
> **Q3: What is the formulation of PU learning? How is it compared to UU learning? This should be introduced before using PU learning as a major topic in L201. PU learning seems to be a special case for UU learning as mentioned in L223.**
>
> **A3:** For the formulation of PU learning (nnPU), see [3]. We have neglected the specific equation of nnPU because it is the most common baseline and widely used in the literature of PU learning, and we have cited it in our paper (see line 196). Also, we stated that PU learning is a special case of UU learning in line 223.
>
> **Q4: I am confused about Sec. 3.2, which is about the motivation of this paper. Does the depicted problem only occur in PU learning? In UU learning? Or both? The author explains the problem by giving PU learning as an example but later talks about UU learning in Sec. 3.3.**
>
> **A4:** Overfitting is a popular phenomenon in the context of weakly supervised learning (WSL), including but not limited to PU learning [3], UU learning [4], noisy label learning [5], Pcomp learning [6], complementary-label learning [7], similarity-confidence learning [8], confidence-difference learning [9], etc. Thus, the problem is not limited to PU learning, even though we use it as an example here. We can observe similar phenomena for many WSL problems.
>
> **Q5: What is CRE $\widehat{R}\_{1}$ in L239? Is it the same as Eq 3, which is $\widehat{R}\_{\rm CUU}$?**
>
> **A5:** The CRE is the most common approach for almost all WSL problems [2-9]. See [2-9] for detailed descriptions and [10] for a systematic overview. Since our goal is to propose a very general framework, there are too many WSL problems with their own CREs, so we did not list them in detail in our paper. However, they can easily be found in related papers. $\widehat{R}_{\rm CUU}$ is a general form for some WSL problems, such as PU learning.
>
> **Q6: In L244, what does it mean "could contain all the unlabeled data whose true labels are not accessible to the learning algorithm"? Is there unlabeled data whose true labels are accessible? How can one access the labels of unlabeled data?**
>
> **A6:** Of course, we cannot access the labels of unlabeled data. However, in some WSL problems, although we do not know the true labels, we can know some weakly supervised information. For example, in UU learning, we know which dataset the unlabeled data comes from. In noisy label learning [5], we know a noisy label that might be true. In complementary-label learning [7], we know a complementary label that specifies the class to which it does not belong. We have changed "unlabeled" to "weakly supervised" for clarity.

---

> ### Author Response · Authors · 2024-11-23
> **Response to Reviewer  JYAz (2/2)**
>
> **Q7: I think the writing of Sec. 3.3 can be improved. Especially, the definition of supervision distillation and model fine-tuning should be explained or summarized first in L237 at the start of the paragraph. Is the first step, i.e. supervision distillation, from L239 to L244, or L239 to L257? What is model-finetuning then?**
>
> **A7:** Supervision distillation refers to obtaining reliable supervision based on the classification head trained by minimizing $\widehat{R}_{1}$, and model fine-tuning refers to performing empirical risk minimization directly using the distilled supervision information. We have revised this paragraph based on your suggestion. Thank you.
>
> **Q8: Are the "class priors" in Sec. 4.3 and 4.4 the same as
> $\pi_{\rm Te}$?**
>
> **A8:** Yes, they are the same.
>
> **Q9: The formulation of Pcomp should be explained somewhere, especially the meaning of "unlabeled pairwise observations were provided" in L402.**
>
> **A9:** Detailed problem settings can be found in [6]. We have cited it in the paper and added some descriptions.
>
> **Q10: The legend in Fig. 2(d) is not visible.**
>
> **A10:** Thanks for your comment and we have fixed the figure.
>
> **Q11: The notation $C$ is first used in Eq. 2. The author also uses $C_g$ and $C_{\ell}$ for various constants in the theoretical analysis. I recommend distinguishing them. Additionally, $D$ is used in Eq. 2 and also as some datasets.**
>
> **A11:** We used $C$ to denote constants and $D$ to denote datasets. The notations mainly refer to similar expressions in previous work of WSL [2-9].
>
> ---
> Reference:
>
> [1] A brief introduction to weakly supervised learning, National Science Review, 2018.
>
> [2] Machine learning from weak supervision: An empirical risk minimization approach, MIT Press, 2022.
>
> [3] Positive-unlabeled learning with non-negative risk estimator, NeurIPS 2017.
>
> [4] Mitigating overfitting in supervised classification from two unlabeled datasets: A consistent risk correction approach, AISTATS 2020.
>
> [5] SIGUA: Forgetting may make learning with noisy labels more robust, ICML 2020.
>
> [6] Pointwise binary classification with pairwise confidence comparisons, ICML 2021.
>
> [7] Learning with complementary labels revisited: The selected-completely-at-random Setting is more practical, ICML 2024.
>
> [8] Learning from similarity-confidence data, ICML 2021.
>
> [9] Binary classification with confidence difference, NeurIPS 2023.
>
> [10] Unified risk analysis for weakly supervised learning, arXiv 2023.

---

> > ### Comment · Reviewer_JYAz · 2024-11-26
> >
> > I thank the authors for addressing most of my concerns, especially for comparing related tasks and modifying Sec. 3. I can see that most of the confusion arises from my not being familiar with the work of literature on PU learning or UU learning. However, I think it indicates a writing problem that this paper is not self-contained and it probably needs more preliminaries or background. Nevertheless, I decide to raise my rating to 5.
> >
> > As for Q11, although they are used in [2-9], I believe it is more important to make notations self-consistent in your paper.

---

> > > ### Author Response · Authors · 2024-11-27
> > > **Thanks for the further feedback!**
> > >
> > > We thank you for your helpful suggestions. We have corrected $C$ and $D$ to $E$ and $F$ in Eq. (2) respectively. Besides, we added a brief introduction to PU and Pcomp learning in Appendix G and the details of UU learning can be found in Section 2.3. We hope that our revision can help you better understand the paper and address your concerns. If you have any further concerns or questions, please do not hesitate to contact us. Thank you again for your help and time in reviewing our submission!

---

### Author Response · Authors · 2024-11-23
**General Response**

We would like to thank the area chairs and reviewers for their efforts and suggestions in reviewing our paper. Their helpful comments have greatly enhanced our work.

We have revised the manuscript according to the reviewers' suggestions and marked them in blue in the new version of the manuscript. The main differences include

- We revised Section 3.2 to present our methodology more clearly.

- We discussed the difference between our work and weakly supervised semantic segmentation in Section 5.

- We included the experiments on ImageNet-100 in Appendix E.

- We included the experiments on CIFAR-100 with a different class partition in Appendix E.

- We added an overall pipeline of WSFT in Appendix A.

---

### Meta-Review · Area_Chair_mfk2 · 2024-12-18

**Metareview:**

This paper explores using pretrained CLIP for weakly supervised learning problem. They shows that CLIP with GPT4o can help surpass existing WSL methods trained from scratch. In addition, they propose Weakly Supervised Fine-Tuning approach with theoretical grounding to effectively fine-tune pre-trained models for various WSL problems.

This paper received mixed reviews, with one positive and three negative scores. The authors are encouraged to improve the formatting for better readability and consider incorporating a broader range of experiments to substantiate their claim that "our proposal can handle any kind of weak supervision generated based on MCD or CCN." At this stage, the paper is not sufficiently prepared for publication.

**Additional Comments On Reviewer Discussion:**

Reviewer JYAz and p8yg raised concerns about the overall readability of the paper. Despite the rebuttal, the methodology and evaluation process remain unclear and require further refinement.

Reviewer tjXA and p8yg expressed doubts about the generalizability of the proposed method to multi-class settings. While the authors included an additional experiment in response, it was insufficient to fully address these concerns.

Reviewer WU3A highlighted the lack of comparisons with recently proposed methods, which may weaken the paper’s positioning.

Overall, this paper requires further refinement and more comprehensive experiments to substantiate its conclusions.

---

### Decision · Program_Chairs · 2025-01-22

Reject